# Ocean internal tides suppress tropical cyclones in the South China Sea

Shoude Guan[1,2], Fei-Fei Jin [3] ✉, Jiwei Tian [1,2] ✉, I-I Lin [4] ✉, Iam-Fei Pun [5], Wei Zhao [1,2], John Huthnance [6], Zhao Xu[1,2], Wenju Cai [1,2,7,8], Zhao Jing [1,2], Lei Zhou [9], Ping Liu[1], Yihan Zhang[1], Zhiwei Zhang [1,2], Chun Zhou [1,2], Qingxuan Yang[1,2], Xiaodong Huang[1,2], Yijun Hou[10] & Jinbao Song[11]

Tropical Cyclones (TCs) are devastating natural disasters. Analyzing four decades of global TC data, here we find that among all global TC-active basins, the South China Sea (SCS) stands out as particularly difficult ocean for TCs to intensify, despite favorable atmosphere and ocean conditions. Over the SCS, TC intensification rate and its probability for a rapid intensification (intensification by $\geq 15.4 \, \mathrm{m \, s^{-1} \, day^{-1}}$) are only 1/2 and 1/3, respectively, of those for the rest of the world ocean. Originating from complex interplays between astronomic tides and the SCS topography, gigantic ocean internal tides interact with TC-generated oceanic near-inertial waves and induce a strong ocean cooling effect, suppressing the TC intensification. Inclusion of this interaction between internal tides and TC in operational weather prediction systems is expected to improve forecast of TC intensity in the SCS and in other regions where strong internal tides are present.

Like tropical cyclones (TCs) over other parts of the ocean, TCs over the South China Sea (SCS) spend most of their lifetime over the sea, and their intensification characteristics have important implications for their consequential impacts on land[1,2]. Timely and accurate forecasts of both TC trajectory and intensity are therefore of great importance for mitigating their damage[1,3]. Nevertheless, despite the rapid improvement in TC track forecast in recent decades, there is much less improvement in TC intensity forecast[4,5]. The intensity of a TC is controlled by its internal dynamics and external ocean and atmosphere conditions in which the TC is embedded[6], including vertical wind shear[7,8], mid-atmosphere relative humidity[9,10], sea surface temperature[11] (SST), and potential intensity[12]. The potential intensity represents the theoretical maximum intensity that a TC can achieve based on the thermodynamic state under given SST and atmospheric

conditions[12,13], while vertical wind shear usually inhibits a TC from reaching its potential intensity[7]. The SST determines the enthalpy flux from the ocean to a TC. When passing over the ocean, the strong winds associated with TCs will entrain subsurface cold water into the surface layer, causing significant SST cooling. This cooling reduces the enthalpy supply to TCs, which is a well-known key negative factor for TC intensification[6,14–17]. Recent studies demonstrated that pre-TC existing oceanic processes, such as mesoscale eddies or barrier layers, can significantly modulate the SST cooling effect and thus influence TC intensification[18–20].

The frequency and intensification of TCs have strong spatial variability over various TC-active oceans[21,22]. The SCS is one of the ocean basins experiencing the most frequent TCs. Previous studies[23–26] on SCS TCs focused on their genesis and interannual

[1]Frontier Science Center for Deep Ocean Multispheres and Earth System (FDOMES) and Physical Oceanography Laboratory/Key Laboratory of Ocean Observation and Information of Hainan Province, Sanya Oceanographic Institution, Ocean University of China, Qingdao/Sanya, China. [2]Laoshan Laboratory, Qingdao, China. [3]Department of Atmospheric Sciences, SOEST, University of Hawaii at Manoa, Honolulu, HI, USA. [4]Department of Atmospheric Sciences, National Taiwan University, Taipei, Taiwan. [5]Graduate Institute of Hydrological and Oceanic Sciences, National Central University, Taoyuan, Taiwan. [6]National Oceanography Centre, Liverpool, UK. [7]Centre for Southern Hemisphere Oceans Research (CSHOR), CSIRO Oceans and Atmosphere, Hobart, TAS, Australia. [8]State Key Laboratory of Marine Environmental Science & College of Ocean and Earth Sciences, Xiamen University, Xiamen, China. [9]School of Oceanography, Shanghai Jiao Tong University, Shanghai, China. [10]Key Laboratory of Ocean Circulation and Waves, Institute of Oceanology, Chinese Academy of Sciences, Qingdao, China. [11]Ocean College, Zhejiang University, Zhoushan, China. ✉ e-mail: jff@hawaii.edu; tianjw@ouc.edu.cn; iilin@ntu.edu.tw

to decadal variability, and found that they are related to the East Asia monsoon, ENSO, Pacific Decadal Oscillation, and so on. To date, few studies examine the TC intensification characteristics in the SCS and related atmospheric and oceanic environments. Recent observational studies demonstrate that SCS background internal tides (ITs), the most powerful in the world ocean[27], could nonlinearly interact with TC-generated ocean near-inertial internal waves[28,29], substantially amplifying turbulent mixing in the upper ocean, potentially affecting the cooling effect and thus modulating TC intensification. However, whether and how ITs affect TC intensification as they pass through the SCS are unknown.

Here, using in situ observations and modeling experiments, we find that the SCS TCs have the globally lowest intensification characteristics despite the high frequency, and show that the ocean ITs in the SCS, mostly due to the Moon-induced astronomic tides flowing over local sharp ocean ridges, are the catalyst for the weak TC intensification.

## Results

### TC intensification in the SCS weakest in the world

Analyzing four decades (1979–2019) of TC data, we examine and compare TC frequency, intensification characteristics and related atmosphere-ocean environmental factors between SCS and global TC-active basins. The TC intensification rate for each TC track-point, which indicates TC intensity change in the subsequent 24 h (in m s$^{-1}$ day$^{-1}$), is calculated and composited in each TC-active basin ("Methods"). The SCS stands out globally as having the weakest intensification characteristics of all TC-active oceans, despite the fact that the SCS is one of the ocean basins suffering the most frequent TCs (Fig. 1). The SCS has the lowest intensification rate, the least chance for rapid intensification (with an intensity increase ≥15.4 m s$^{-1}$ day$^{-1}$), and the lowest percentage of intense (Category 3–5) TCs (Fig. 1 and Table S1; "Methods"). Specifically, the TC intensification rate for the SCS and global average is 1.4 and 2.6 m s$^{-1}$ day$^{-1}$, respectively, while the percentage of intense TCs is ~3% for the SCS compared to ~11% for the global average (Fig. 1 and Table S1). The long-term average probability

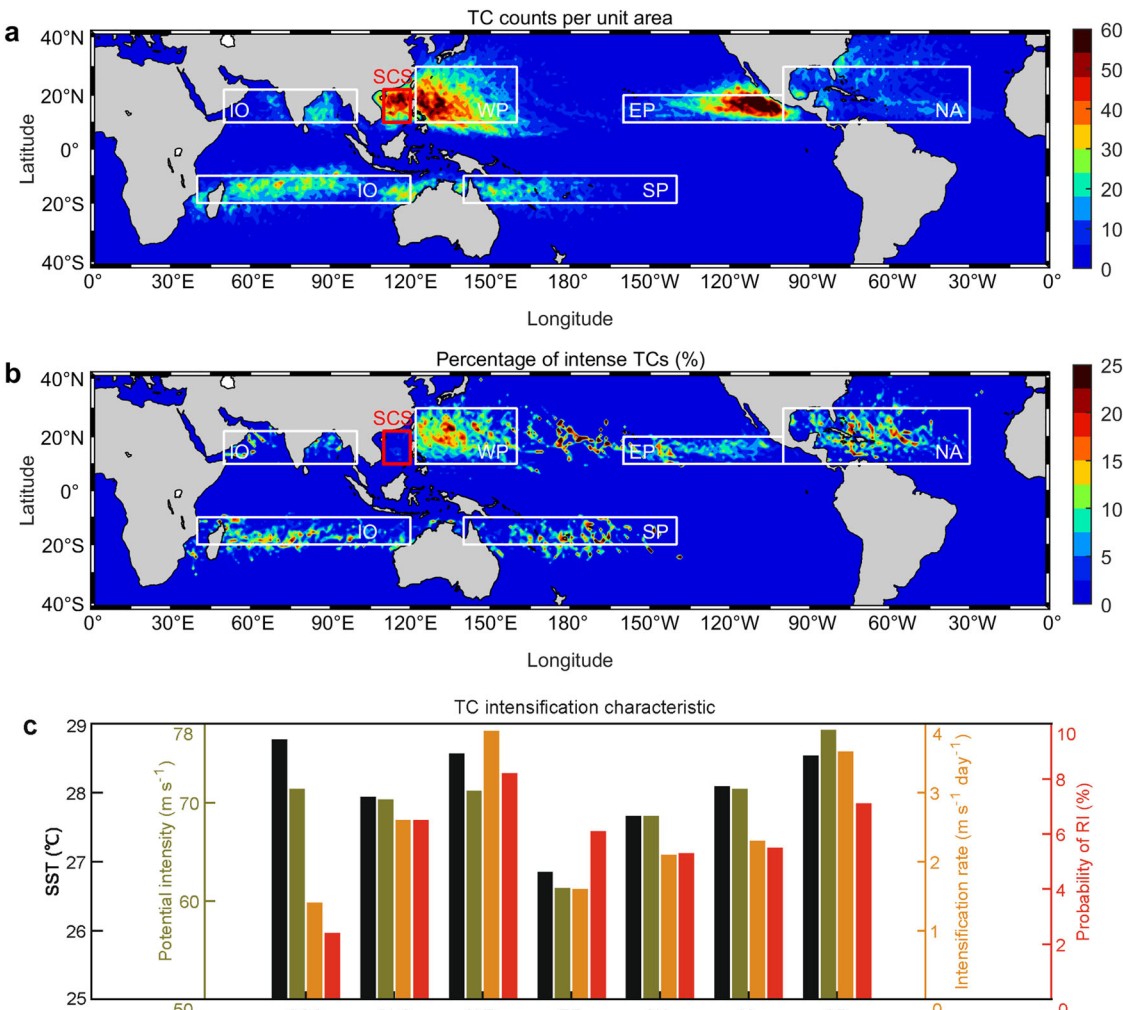

**Fig. 1 | Global distribution of tropical cyclone (TC) occurrences and environmental factors. a** TC track-point counts (6 hourly, from Tropical Depression to Category 5 TC in the Saffir-Simpson Scale) during the 1979–2019 period, in 1° by 1° grid. **b** As in (**a**), but for the percentage of intense TCs (Category 3–5 counts over all counts) in 1° by 1° grid. **c** Area-averaged climatological sea surface temperature (SST, °C, black), potential intensity (m s$^{-1}$, olive), intensification rate (m s$^{-1}$ day$^{-1}$, orange), and probability of rapid intensification (RI (%), number of rapid intensification TC track-points over all TC track-points, red) of the six TC-active oceans. The TC-active oceans include the South China Sea (SCS), the Western North Pacific

Ocean (WP), the North Atlantic Ocean (NA), the Eastern North Pacific Ocean (EP), the Indian Ocean (IO), and the South Pacific Ocean (SP). GLO means the global average over the other five TC-active ocean except the SCS. The Indian Ocean (IO) here combines the north and south Indian oceans. During the 1979–2019 period, the SCS features the most frequent TCs but the lowest percentage of intense TCs, the lowest probability of rapid intensification and the lowest intensification rate among global TC-active oceans, despite its highest sea surface temperature and 2nd highest potential intensity. Source data are provided with this paper.

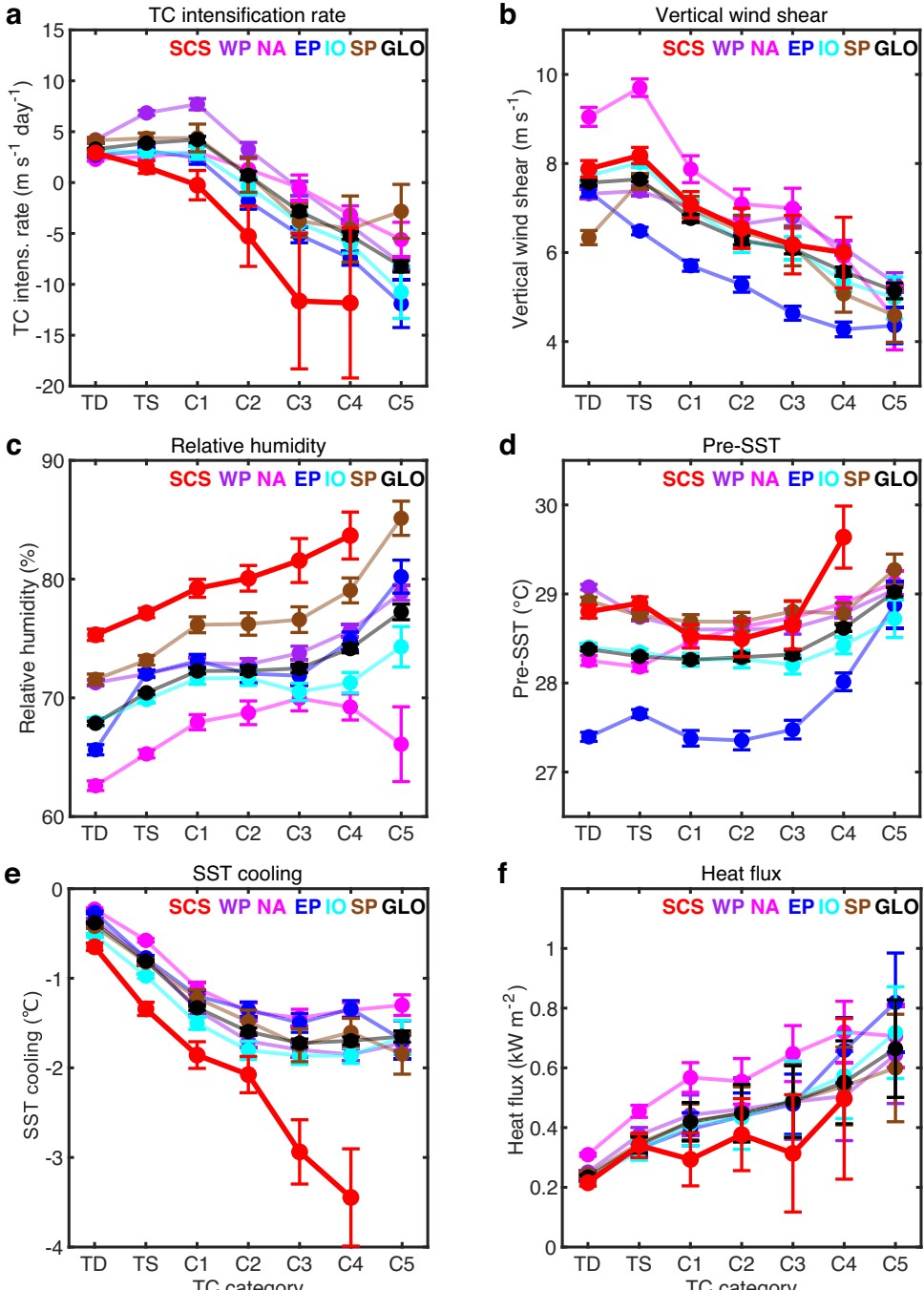

**Fig. 2 | Composites of tropical cyclone (TC) intensity controlling factors in TC-active oceans. a** TC intensification rate (m s⁻¹ day⁻¹) versus TC category.
**b** Atmospheric vertical wind shear (m s⁻¹) between 200 and 850 hPa during TCs versus TC category. **c** Relative humidity (%) averaged between 500 and 700 hPa within 500 km of the TC center versus TC category. **d** Pre-TC sea surface temperature (Pre-SST, °C) versus TC category. **e** TC-induced SST cooling effect (°C) versus TC category. **f** Surface heat flux supply (sensible heat + latent heat) (kW m⁻²) during TCs versus TC category. Error bars in (**a**–**e**) indicate the 90% confidence interval. The error bar in (**f**) is derived based on a factor ranging from 0.3–0.7 when

estimating during-TC SST cooling based on satellite observations ("Methods"). TC-active oceans include the South China Sea (SCS), the Western North Pacific Ocean (WP), the North Atlantic Ocean (NA), the Eastern North Pacific Ocean (EP), the Indian Ocean (IO), and the South Pacific Ocean (SP). GLO means the global average over the five TC-active basins except the SCS. The lowest intensification rate in the SCS results from the strongest TC cooling effect and the lowest heat flux among global TC-active oceans. Other atmospheric and oceanic factors influencing TC intensity are favorable or moderate for TC intensification in the SCS. Source data are provided with this paper.

of rapid intensification is only ~2%, compared to ~7% probability in all other oceans. Considering the dependence of intensification rate on TC intensity[30], we further compare the intensification rate at the same TC intensities, which shows that the intensification rate in the SCS is always the lowest among global TC-active oceans (Fig. 2a).

The weak intensification characteristics in the SCS are at odds with the relatively favorable atmospheric and oceanic environmental conditions therein. Using 41 years of global atmospheric and oceanic datasets, we calculate and composite related environmental factors, such as SST, potential intensity, vertical wind shear, and relative

humidity. We find that none of these factors explain the unusually weak intensification characteristics of SCS TCs. Specifically, the SCS has a relatively high SST in global oceans, which is obviously not an intensification-limiting factor; the SCS's potential intensity is the 2nd highest in the world; the vertical wind shear in the SCS is not substantially different from most other oceans; the SCS's mean relative humidity is the highest and should be conducive to intensification (Figs. 1c and 2 and Table S1).

### Small SCS basin size and landmass geography not responsible

In the SCS, the small ocean basin and surrounding landmass (or islands) effect are conventionally considered as candidates for the lack of intense TCs[31]. However, after excluding these two factors, the intensification characteristic of SCS TCs is still the weakest. For instance, although SCS basin size is smaller than the other five TC-active oceans, intensification rate and probability of rapid intensification have little association with the overall basin size. The percentage of intense TCs could be affected by the small basin size, but when using a fixed-window analysis ("Methods"), which compares the percentages calculated when TCs in all oceans travel as long a distance as those in SCS, the SCS percentage of intense TCs is still the lowest (Fig. S1).

To rule out the landmass effect in the SCS, TC track-points within 100 km of the land or landfall within 24 h have been pre-excluded in the calculation of TC intensification rate[32]. The landmass associated with the high mountain in Philippines also has little impact on intensification rates of SCS TCs, as indicated by the consistently low intensification rates regardless of whether TCs pass over Philippine-mountain or not (Fig. S2). Therefore, neither the small basin size and landmass effect nor the known atmosphere-ocean factors can account for the weakest intensification characteristic of SCS TCs.

### Unusually large cooling effect in the SCS

Another well-known key negative factor for TC intensification is the TC self-induced oceanic cooling effect[15]. Though pre-TC SST is warm, strong TC-ocean coupling effect can cool the ocean quickly, reduce the ocean's enthalpy supply for TCs, and thus suppress TC intensification[32–34]. We therefore examine the possible contribution from the TC self-induced ocean cooling effect. We find that the cooling effect of the SCS TCs is indeed the strongest among all global TC-active oceans (Fig. 2e) and the corresponding enthalpy flux that is thus the least (Fig. 2f). In particular, cooling and enthalpy flux in the SCS are ~80% stronger and ~33% less, respectively, than the global averages for intense TCs (Table S1). Thus, despite the highest pre-TC SST of all oceans, SCS SST drops most rapidly as TCs pass through (Fig. S3); air-sea enthalpy flux from the ocean to a TC is therefore substantially reduced and TC intensification is curtailed. A key question is what causes such extraordinary cooling effect in the SCS that is much stronger than in other oceans.

The cooling effect is a function of pre-TC upper ocean condition and TC attributes, increasing with pre-TC upper ocean thermal stratification, TC intensity, TC size, and decreasing with TC translation speed (Uh) (refs. 15,34); however, none of these accounts for the large cooling effect in the SCS. Specifically, all TC attributes such as average intensity, size, and Uh, in the SCS are moderate or indistinguishable from the global average (Fig. 3a, b and Table S1), unable to induce the strongest cooling. The SCS ocean subsurface thermal stratification is strong[35,36] but is smaller than that in the Eastern Pacific (Figs. 3c, d and S4), yet despite the weaker stratification, SCS TC cooling effect is 50–100% larger than that in the Eastern Pacific for TCs of the same TC intensity, size, and translation speed (Figs. 2e and S5). Thus, nor can stratification explain the observed disproportionately strong cooling effect in the SCS.

### ITs drive the unusual cooling effect

The above analysis suggests a unique process contributing to the unusual cooling effect that operates in the SCS. Given that the SCS hosts the world's most powerful Moon-induced ITs[27], which are located en route of the frequent TC passages (Fig. S6), we explore their potential impacts. The SCS ITs are characterized by diurnal (D1, once-daily) and semidiurnal (D2, twice-daily) frequencies, generated through interaction between the sharp ocean ridges and strong astronomic tides at the Luzon Strait. Upon generation, they propagate into the SCS and dissipate on the continental shelf[27,37]. Previous observational studies[28,29] reported that ITs could linearly and non-linearly interact with TC-generated ocean near-inertial internal waves, substantially amplifying turbulent mixing in the upper ocean. Because TC cooling effect is dominated by the turbulent mixing originating from the strong vertical shear of the horizontal currents in TC-generated near-inertial internal waves[15], the quasi-permanent presence of the SCS powerful ITs provides a highly dynamic background to boost the cooling effect via interaction between background ITs and TC-generated near-inertial internal waves[28,29].

We searched in situ ocean current observations from global mooring and buoy databases to compare the ocean responses to TCs under similar TC conditions, but with/without background ITs ("Methods"). This search was a non-trivial task because TCs can be captured only when they pass over the mooring sites. The two companion TC cases for comparison must also be of similar TC attributes. Furthermore, the mooring sites need to have subsurface current measurements at different depths. After an extensive search, we identified two comparable TC cases, one captured by a mooring in the SCS (with ITs) from the South China Sea Mooring Array (SCSMA), and the other captured by a buoy in the North Atlantic (NA; without ITs) (Fig. S7), both of ~24 m s$^{-1}$ maximum sustained wind speed and ~4–5 m s$^{-1}$ Uh. The mooring and the buoy were located 120 km and 110 km to the right of the TC track, respectively. Both the mooring and buoy were located out of eddy regimes, indicating negligible influence on the comparison from background geostrophic currents (Fig. S8).

In the NA case, the near-inertial current structure dominates, with a maximum amplitude of ~0.4 m s$^{-1}$ (Fig. 4a), which is consistent with existing literature[15,38]. As expected, the energy spectrum of upper ocean currents and their vertical shear were concentrated at the local inertial frequency ($f$) (Figs. 4b and S7). However, in the SCS case, ocean currents showed additional spectral peaks from different waves compared to the NA case, with a much larger amplitude, i.e., up to 0.8 m s$^{-1}$ (Fig. 4b, c). TC-generated near-inertial waves and the background diurnal and semidiurnal ITs are all present during the SCS TC passage. In addition, small-scale secondary waves ($fD1$, $fD2$, $D2f$) ("Methods"), induced by nonlinear coupling between TC-generated near-inertial internal waves and background ITs[28,29], are found in the energy spectrum[39,40]. Hence, the total wave energy and shear variance are approximately seven times larger in the SCS than those in the NA case. Consequently, the inverse Richardson number, which measures the strength of turbulent mixing[41], is much larger in the SCS case, facilitating stronger subsurface entrainment that leads to stronger cooling by ~0.5–1.0 °C than that in the NA, despite the fact that SCS's subsurface thermal stratification was relatively weaker (Figs. 4d and S7).

### Cooling by ITs confirmed in model experiments

That the SCS ITs are the catalyst for the extraordinary TC cooling effect is further confirmed by numerical experiments, using two different but complementary models. The first model is Price model[42], which is a simple model designed for estimating the TC-induced cooling effect ("Methods"). Due to its efficiency, it has the advantage for simulating a large number of TC cases. Here 19 years of SCS TCs (1998–2016, 661 6-hourly TC track points) were simulated for the scenarios without and with ITs separately, by incorporating a typical SCS ITs current profile

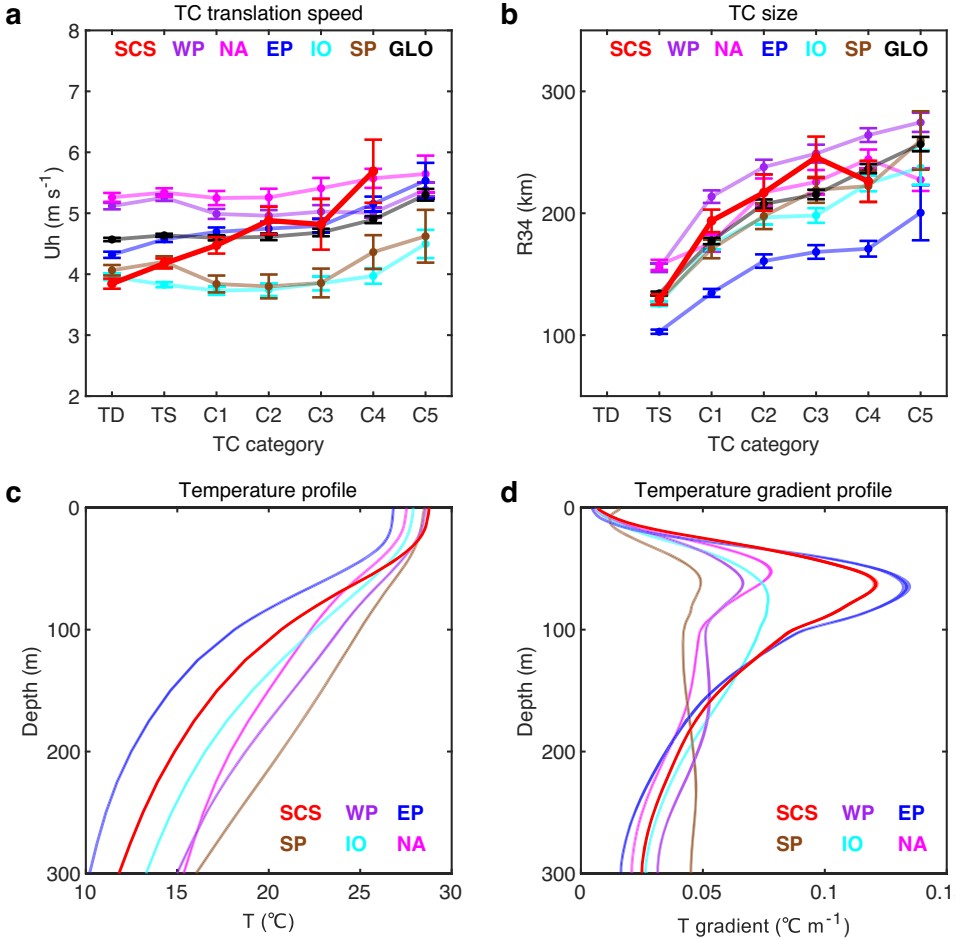

**Fig. 3 | Comparison of factors affecting tropical cyclone-induced (TC-induced) sea surface temperature (SST) cooling among TC-active oceans. a** Average TC translation speed (Uh, m s$^{-1}$) at each TC category. The translation speed is calculated by dividing the distance the storm moves 6 h prior to and 6 h after reaching the current position by the total time interval (12 h). **b** Average TC size (km) at each TC category. TC size here is characterized in terms of the maximum radial extent of 34 kt wind speed (R34), obtained by averaging the R34 values of four compass quadrants from the IBTrACS global TC best track dataset. Error bars in (**a**, **b**) indicate the 90% confidence interval. **c** Area-averaged climatological temperature profiles (T) derived from the WOA13 dataset. **d** Area-averaged climatological vertical temperature gradient ($\frac{\partial T}{\partial z}$) profiles from the WOA13 dataset. The shading in (**c**, **d**) indicates the 99% confidence interval. TC-active oceans include the South China Sea (SCS), the Western North Pacific Ocean (WP), the North Atlantic Ocean (NA), the Eastern North Pacific Ocean (EP), the Indian Ocean (IO), and the South Pacific Ocean (SP). GLO means the global average over the five TC-active basins except the SCS. TC Uh and size in the SCS are indistinguishable from other TC-active oceans. The ocean subsurface thermal stratification in the SCS is strong but is weaker than that in the Eastern Pacific. Thus, neither TC attributes nor stratification can explain the observed strong cooling effect in the SCS. Source data are provided with this paper.

from in situ observations into the model (Fig. S9). The second model is the three-dimensional full ocean model ROMS (Regional Ocean Modeling System) and its advantage is for a detailed case study[43]. Here we simulate the case of TC Megi in 2010 (Fig. S10a). The Megi case was further coupled to the Atmospheric WRF (Weather Research and Forecasting) model to examine the effect of ITs on TC intensity ("Methods").

The results from the Price model experiment show that without ITs, SST cooling is underestimated by ~34%, at 1.64 °C compared to the observed averaged cooling of 2.47 °C (Fig. 5a); with ITs the cooling is 2.38 °C, much closer to the observations. The in situ shear profile of the ITs features the largest shear at ~70–100 m depth (Fig. S9), which is also the known depth for intense TC-ocean interaction[15,42]. In other words, the vertical structure of the SCS ITs locates its shear at an optimal depth favoring TC-ocean interaction. Using the 34% less cooling, we estimated the air-sea enthalpy flux to TCs for the without-ITs scenario. If without ITs, the enthalpy flux in the SCS would be comparable with or even somewhat larger than global average (Fig. 5b). Nevertheless, due to the presence of ITs, the enthalpy flux is

the lowest in the world (Figs. 5b and 2f). Results from the ROMS model experiments for TC Megi (2010) confirm the effect of ITs to promote the extraordinary TC cooling effect in the SCS. The strength of subsurface turbulent mixing in the with-ITs simulation is much larger than that in the without-ITs simulation (Fig. 5c), enabling more cold water to be entrained into the sea surface in the presence of ITs (Fig. 5d). When coupling with WRF model, TC Megi's intensity and associated rainfall are both obviously weakened in the with-ITs simulation (Figs. S11 and S12).

## Discussion

Our study here reveals a dynamic process of how the ocean subsurface ITs can affect weather phenomenon, as manifested in the suppression of TC intensity in the SCS. Through complex linear and non-linear interactions between ITs and SCS TCs, unusually large TC self-induced cooling effect is induced to suppress TC intensification. Considering multiple sources and long-range propagation of ITs, the strength of ITs in the SCS is spatially inhomogeneous[44,45] and thus potentially influences the suppression effect. For instance, we contrasted the

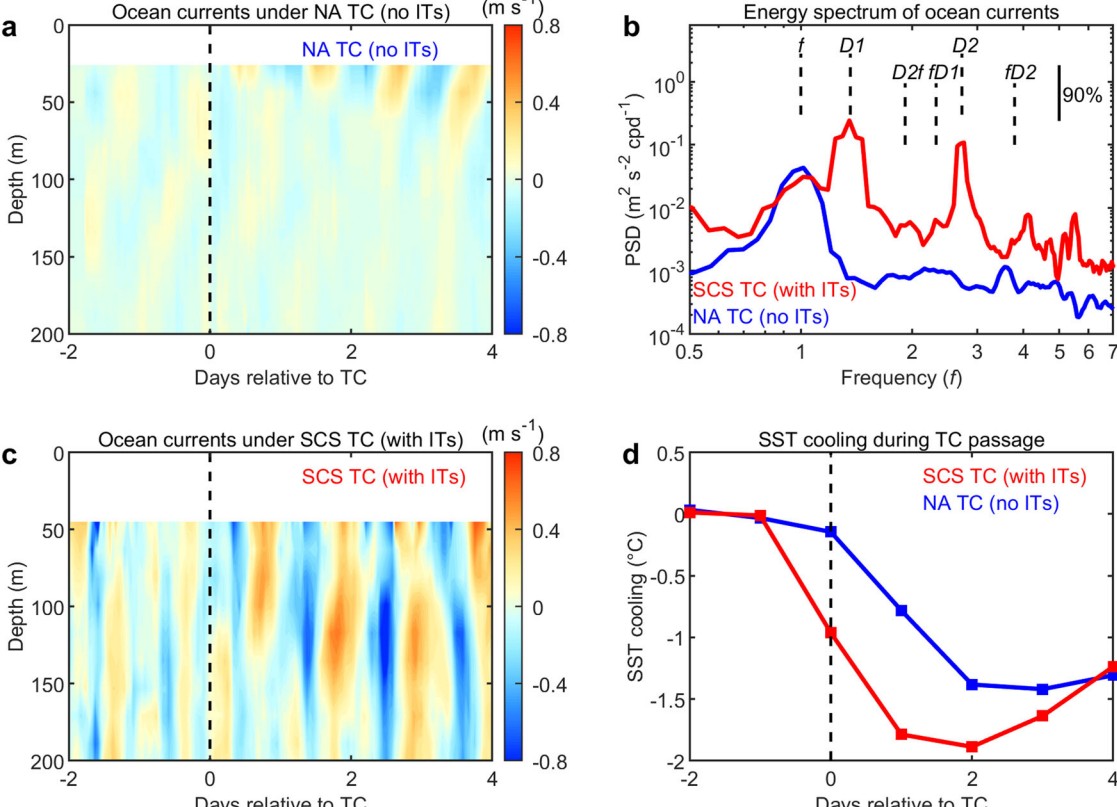

**Fig. 4 | Observed tropical cyclone-ocean (TC-ocean) interactions with and without background internal tides (ITs). a** Buoy-observed ocean horizontal currents during the passage of the North Atlantic (NA) TC (no ITs). **b** Power spectrum density (PSD) of horizontal currents during and after TCs with (red) and without (blue) ITs. The PSD is averaged over the upper 150 m. The vertical solid line at the upper right corner indicates the 90% confidence interval. The frequency of near-inertial internal waves ($f$), diurnal (D1) and semidiurnal (D2) ITs, and the secondary waves ($fD1, fD2, D2f$) generated by nonlinear coupling between near-inertial internal waves and ITs are indicated by vertical dashed lines. The local near-inertial frequency depends on the latitude where the mooring or buoy is located. At the observation positions in the South China Sea (SCS) and the NA, near-inertial frequencies ($f$) are 0.71 cpd (cycles per day; equals to ~34 h period) and 0.93 cpd (~26 h period), respectively. Thus, the x-axis is scaled to their own local inertial frequencies ($f$). **c** Mooring-observed ocean horizontal currents during the passage of the SCS TC (with ITs). **d** Satellite-observed sea surface temperature (SST) cooling at the mooring/buoy sites. The vertical dashed lines in (**a**, **c**, **d**) indicate the time of TC passage. Under similar TC conditions, the in situ observations indicate background ITs in the SCS amplify the subsurface entrainment and enhance SST cooling. Source data are provided with this paper.

ITs-enhanced TC-ocean cooling effect when TCs passed over the north or south SCS. From two mooring observations, the strength of ITs in the north SCS is about six times larger than that in the south SCS (Fig. S13), implying more ITs' energy supply to enhance the TC-ocean cooling effect. The SST cooling difference between north and south SCS for intense TCs is about 0.5 °C. Correspondingly, the amplitude of negative intensification rate in the north SCS (−13.2 m s⁻¹ day⁻¹) was 47% larger than that in the south SCS (−9.0 m s⁻¹ day⁻¹) for intense TCs. The monsoon flow can also potentially influence the seasonal variation of internal tidal energy flux by modulating ocean circulation and stratification[46,47]. The total energy flux of SCS ITs peaks in summer and autumn (Fig. S6), which is temporally consistent with the peak season of typhoons in the SCS[23], potentially heightening the suppressing effects of ITs on SCS TCs. Overall, how best to incorporate these complex processes in coupled numerical models for TC forecasting operationally is thus an important next step forward.

For a long time, ITs hidden below the sea surface appeared irrelevant to people's life on land, and were thought to only impact marine activities such as underwater navigation, offshore drilling, marine biogeochemistry and earth climate[48–50]. Here we demonstrate that these powerful ITs are highly relevant to people's livelihood on land. In fact, they are the unsung heroes to the half-billion people living in the Asian coast, for they effectively suppress SCS TCs' intensification.

Throughout human history, folktales fantasized about Moon-to-weather connections. The dynamic process reported here serves as one credible scientific linkage between the oceanic tides owing to the Moon and the most devastating weather phenomenon, TCs. Another similar ocean basin probably subject to the suppressing effect of ITs on TCs, is the Coral Sea of the South Pacific which also features strong ITs radiated from surrounding submarine ridges and straits[51,52]. As expected for intense TCs, the average SST cooling at −2.2 °C, TC intensification rate at −11.0 m s⁻¹ day⁻¹, and percentage of intense TCs at ~4.3%, are all substantially lower than global average. Despite the weak TC intensification characteristics and enhanced cooling effect as that in the SCS, the potential effect of ITs on TCs in the Coral sea is yet to be examined.

Impacts of ITs may not be limited to intense TCs, but extend to weaker TCs such as Tropical Storms/Depressions (Fig. 2e). This suggests that other weather phenomena may be buffered by ITs. Given ITs are present in many other oceans, e.g., the Hawaiian ridge[53], the Indonesian seas[54], and the British shelf seas[55], our study thus opens up a field to explore the ITs-weather connection. How ITs in other oceans interact with a broad spectrum of weather phenomena, e.g., monsoon, frontal systems, Maddan-Julian Oscillations[46] and the consequential impact on people's well-being will be scientifically stimulating research that is of societal benefit.

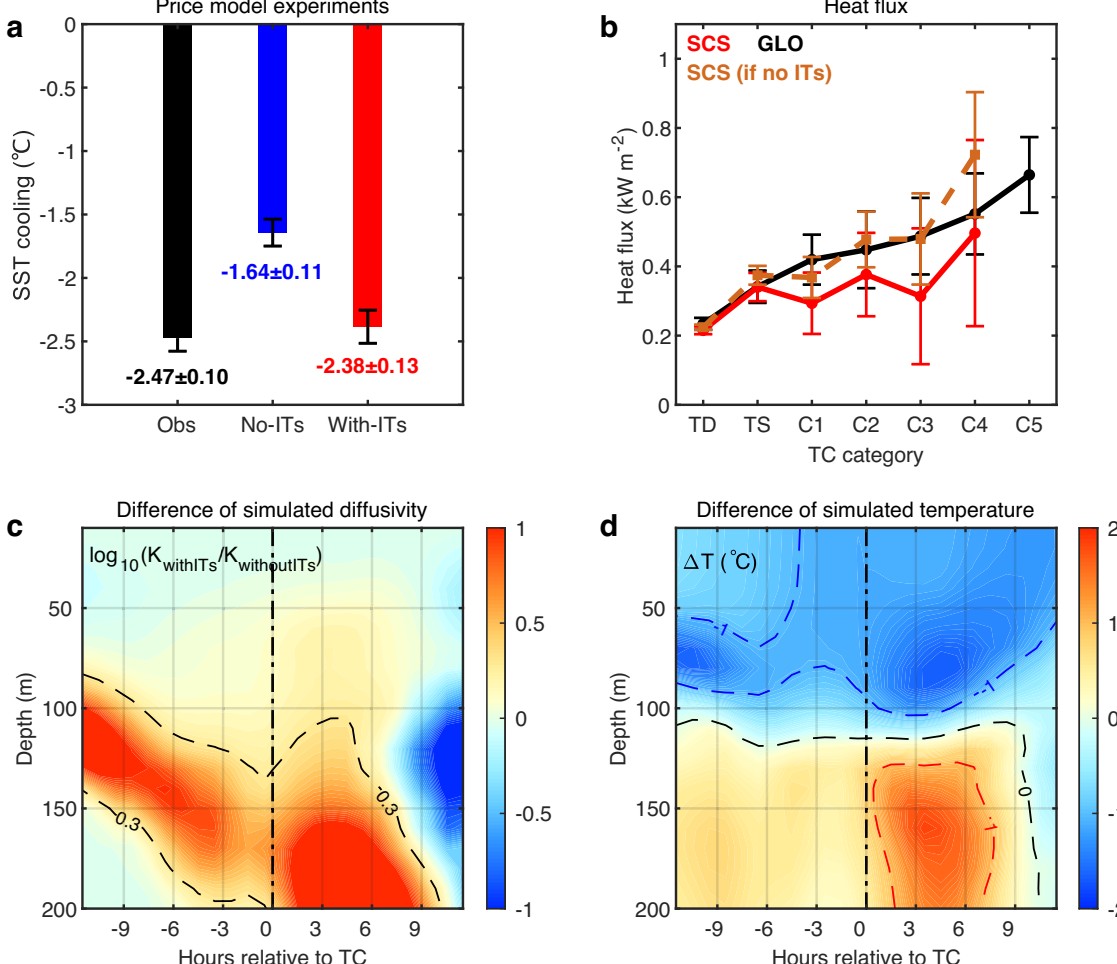

**Fig. 5 | Model simulated tropical cyclone-ocean (TC-ocean) interactions with and without background internal tides (ITs). a** Satellite-observed (Obs, black) and Price model simulated sea surface temperature (SST) cooling for the scenarios with (red) and without (blue) background ITs for TC Megi in 2010. Error bars indicate the 99.99% confidence interval. The large SST cooling from satellite observation is simulated in the with-ITs simulations but not in the without-ITs simulations, and the simulated difference is significant, based on the Student's *t* test. There is no significant difference between Obs and with-ITs simulations. **b** Air-sea enthalpy (sensible heat + latent heat) fluxes (kW m$^{-2}$) for the hypothetical scenario if the South China Sea (SCS) has no ITs (dashed-chocolate curve). For reference, the composited enthalpy fluxes for the SCS and global average (GLO) based on observations (same as in Fig. 2f) are also shown. Error bars indicate the 90% confidence interval. **c** Difference of ROMS model simulated temperature diffusivity between with-ITs and without-ITs simulations. The background color shows $\log10$ ($K_{withITs}/K_{withoutITs}$), where $K_{withITs}$ and $K_{withoutITs}$ are the temperature diffusivity coefficients outputted by simulations of with-ITs and without-ITs, respectively. **d** As in (**c**), but for temperature difference (temperature of with-ITs simulation minus that of without-ITs simulation). Blue, black and red dashed lines indicate contours of −1.0 °C, 0.0 °C and 1.0 °C, respectively. The results are averaged along the 21°N section indicated in Fig. S10a. In the with-ITs simulation, upper ocean mixing is heightened by 2–10 times, resulting in enhanced SST cooling and a consequently upper-layer cooler (above 100 m) and sub-layer warmer (below 100 m) pattern in the temperature difference. Both model experiments indicate that the background ITs are the catalyst for the extraordinary TC cooling effect in the SCS. Source data are provided with this paper.

## Methods

### Data
Global TC best track datasets from 1979 to 2019 (including 6-hourly TC center position, 1-min sustained maximum wind speed, minimum central pressure, radius of 34 kt wind speed), distributed by the US Joint Typhoon Warning Center (JTWC) and the National Hurricane Center (NHC), are obtained from the website of the International Best Track Archive for Climate Stewardship (IBTrACS)[56]. World Ocean Atlas 2013 (WOA13) dataset is provided by the NOAA National Centers for Environmental Information[57].

Due to a cloud-penetrating capability[58], satellite microwave SST measurements provide reliable estimates of SST cooling induced by TCs, and have been widely used in existing literature[32,34,59–61]. The daily microwave SST data (from 1998 to 2019) used to composite pre-TC SST and TC-induced SST cooling in

this study are obtained from the Remote Sensing Systems (RSS), which is sponsored by the NASA Earth Science REASNN DISCOVER Project and provides optimally interpolated daily microwave SST products since 1998.

Atmospheric wind, temperature, and humidity profile datasets used to estimate the TC potential intensity and vertical wind shear are obtained from the European Centre for Medium-Range Weather Forecasts (ECMWF) Interim Reanalysis database.

To contrast the upper ocean response under similar TC conditions, but with or without background ITs, mooring observations in the South China Sea and buoy observations in the North Atlantic Ocean are searched. The buoy data are available from the National Data Buoy Center (NDBC). The mooring data are from the South China Sea Mooring Array (SCSMA) constructed by the Ocean University of China (see details below).

## SCS and other five tropical TC-active oceans, and their cyclone intensity properties

TC-active oceans are defined as follows: South China Sea (SCS): 10°–22°N, 110°–120°E; Western North Pacific (WP): 10°–30°N, 122°–160°E; Eastern North Pacific (EP): 10°–20°N, 100°–160°W; North Atlantic (NA): 10°–30°N, 30°–100°W; Indian Oceans (IO) includes the north Indian ocean (10°–22°N, 50°–100°E) and south Indian ocean (10°–20°S, 40°–120°W); South Pacific (SP): 10°–20°S, 140°E–140°W. Modifying the domain range of each ocean does not materially affect our results and conclusions. In our study, the TC season is defined as June–October in the Northern Hemisphere and November–March in the Southern Hemisphere, but for the north Indian Ocean (0°–25°N, 30°–100°E), TC season is defined as April–May and September–November[62].

TC intensity, measured as its 1-min maximum sustained wind speed (*Vmax*), follows the Saffir-Simpson scale: Tropical Depression (TD; *Vmax* < 18 m s⁻¹), Tropical Storm (TS; 18 m s⁻¹ < *Vmax* < 33 m s⁻¹), Category 1 (C1; 33 m s⁻¹ < *Vmax* < 43 m s⁻¹), Category 2 (C2; 43 m s⁻¹ < *Vmax* < 50 m s⁻¹), Category 3 (C3; 50 m s⁻¹ < *Vmax* < 58 m s⁻¹), Category 4 (C4; 58 m s⁻¹ < *Vmax* < 69 m s⁻¹), and Category 5 (C5; *Vmax* > 69 m/s).

Rapid intensification of TC is defined as an increase in TC's maximum sustained wind speed of ≥30 kts (15.4 m s⁻¹) over a 24-h period[63]. In the present study, two methods, i.e., TC-case (over the TC's lifetime as a case) based and the TC track-point (6-hourly) based, are employed to estimate the probability of rapid intensification, using the TC best track dataset in the recent 41 years (1979–2019)[56]. For the TC-case-based method, probability of rapid intensification is obtained as the number of TC cases undergoing at least one rapid intensification event over all TC cases when passing each specific basin. For the TC track-point-based method, probability of rapid intensification is obtained as the number of TC track-points undergoing rapid intensification events (only the first track-point is accounted for one rapid intensification event) over all TC track-points passing the specific basin during 1979–2019. Results of both methods are shown in Table S1, consistently indicating that the SCS has the lowest chance for rapid intensification among global TC-active oceans. For simplicity, only the result from the TC track-point-based method is shown in the main text and Fig. 1c.

Percentage of intense (Category 3–5) TCs is also calculated based on the two methods as for rapid intensification probability. For the TC-case-based method, it is defined as the number of TC cases reaching Category 3–5 over the total TC cases when passing each specific basin in recent 41 years (1979–2019). For the TC track-point-based method, the intense TC percentage is defined as number of Category 3–5 TC-track points over the total track-points in recent 41 years (1979–2019). For the intense TC, percentage in Fig. 1b is calculated via the TC track-point-based method, but in each 1° by 1° grid. Results of both methods are shown in Table S1, consistently indicating that the SCS has the lowest percentage of intense TCs among global TC-active oceans. For simplicity, as for the probability of rapid intensification, only the result from the TC track-point-based method was shown in the main text and Fig. 1b.

TC intensification rate (in m s⁻¹ day⁻¹) for each TC track-point is calculated as the linear regression coefficient of the maximum sustained wind speed over five data points (i.e., 24 h)[18], which include the current and the four subsequent six-hourly TC points. TC track-points within 100 km of the land or landfall within 24 h are excluded to avoid land effects[32]. In Fig. 2a, the composited intensification rate in the recent 41 years (1979–2019) was obtained by averaging all values at each category bin within each TC-active ocean. Approaches to calculating intensification rate used in other studies[64], for example, by directly differencing the maximum wind speeds at 6-h, 12-h, 18-h or 24-h intervals, or by a linear regression of three to four TC track-points (12-h or 18-h intervals), are also tested and all approaches show that the

SCS had the lowest intensification rate among the six TC-active oceans (not shown here). There are only two TC track-points of Category 5 in the SCS during the study period, but the two track-points are too close to the land, hence not considered here.

## Factors controlling TC intensification

Impact of basin size is examined using a fixed-window analysis. Because the relatively small basin size of the SCS may impact the percentage of intense TC results, we conduct a fixed-window analysis to compare the situation that if all TCs traveled the same distance as SCS TCs. During the study period, there were 150 TCs generated locally in the SCS, and the average traveling distance over ocean (i.e., from the genesis point to the landfall point) was ~1518 km. We then extract TC segments from genesis to 700–1518 km for all TC cases in other TC basins, so that all basins are now compared under the same spatial window. The results show that SCS has the lowest percentage of intense TCs in all segments (Fig. S1). These suggest that something beyond the basin-size factor must operate to hinder TC intensification in the SCS.

Potential intensity theoretically estimates the upper bound of TC's intensity[65]. We base our potential intensity calculations on a Fortran program (https://emanuel.mit.edu/products) provided by K. Emanuel (MIT, USA) via his website[12], using atmospheric input from the monthly atmospheric temperature and humidity profile of the ECMWF interim database, and SST input from the monthly ocean temperature from the WOA13.

Vertical wind shear is calculated as the 200 hPa and 850 hPa wind difference, averaged within 500 km of the TC center[66]. We obtained the six hourly wind fields at 200 hPa and 850 hPa from the ECMWF interim database at each 0.25-degree grid during 1979–2019.

Pre-TC SST and TC-induced SST cooling are computed using daily satellite microwave SSTs at each TC track-point. For each TC track-point, the pre-TC SST is obtained via averaging SSTs between 1–7 days before TC occurrence (i.e., day −7 to day −1), within 100 km of the TC center. The TC-induced SST cooling is defined as the mean SST between day 0 to day 1 minus the pre-TC SST. The results in Fig. 2d, e are the averaged values at each category bin during the 1998–2019 period, since the satellite microwave SST datasets are available from 1998. To ensure robustness, we also test using various other definitions, for example, using 50–200 km average SST cooling within the TC center or the SST changes between days 0 to 5 (during and after TCs) and days −10 to −1 (before TCs). The results remain consistent. Figure S3 further examines the temporal evolution of TC-induced SST cooling and indicates that SST cooling in the SCS is nearly always larger than that in the other five oceans during and after TC passage.

Total surface enthalpy heat flux, i.e., sensible heat ($Q_s$) plus latent heat ($Q_L$) flux, from the warm ocean to a TC is estimated using the bulk aerodynamic formulae[67,68]:

$$Q_S = C_H V (T_s - T_a) \rho_a C_{pa} \qquad (1)$$

$$Q_L = C_E V (q_s - q_a) \rho_a L_{va} \qquad (2)$$

where $C_H$ and $C_E$ are exchange coefficients of the sensible and latent heat, respectively; $V$ is TC wind speed; $T_s$ and $T_a$ are during-TC SST and near-surface air temperature, respectively; $q_s$ and $q_a$ are surface and air specific humidities under TC, respectively; and $\rho_a$, $C_{pa}$, and $L_{va}$ are air density, heat capacity of the air, and latent heat of vaporization, respectively. For each TC track point, $T_s$ is primarily based on the satellite microwave-observed SST. $T_a$ is obtained from ECMWF Reanalysis dataset. $q_s$ and $q_a$ are computed from $T_s$ and dew point temperature, obtained from the ECMWF Reanalysis dataset. Both $C_H$ and $C_E$ were taken as $1.3 \times 10^{-3}$, based on observations under TC

conditions[69]. The satellite microwave SST usually overestimates the SST cooling under the TC core[70], due to possible rain effect[58]. For flux calculation, during-TC SST is estimated based on pre-TC SST minus satellite-derived cooling multiplied by a factor of 0.5 with a range between 0.3 and 0.7 (error bars in Fig. 2f). This 0.5 cooling factor is applied, based on that the during-TC cooling is smaller than the cooling soon after TC's passing[70]. Because large variability also exists, the 0.5 factor is used here[70].

## Impact of internal tides on upper ocean dynamical response to TCs

Mooring observations in the SCS used in this study (Figs. 4 and S7) are obtained from the SCSMA (South China Sea Mooring Array), which is designed and constructed by the Ocean University of China since 2009, to investigate the spatiotemporal characteristics and underlying dynamics of multiscale dynamic processes including the internal waves, meso- to submesoscale eddies, and deep circulation in the SCS[28,71]. The SCSMA maintains more than 40 moorings in the SCS annually and generally recovers and deploys the moorings once a year mostly in the boreal spring or summer. Most of these moorings were equipped with an upward-looking 75 kHz Acoustic Doppler Current Profiler (ADCP; sometimes with an additional downward-looking ADCP) at about 500 m depth to measure ocean current velocity profiles continuously in the upper 500 m (or 1000 m) water column. In other TC-active ocean basins, available mooring or buoy observations with subsurface current information during TC passages are very rare. To the best of our ability, we obtained three buoys (42002, 42038, and 42041) with upper ocean current observations in the NA (without ITs) from the NDBC.

To contrast the upper ocean dynamical responses to TCs with or without background ITs, ocean current observations under two similar TC cases, one in the SCS (with strong ITs) and one in the NA (without ITs), are compared (Figs. 4 and S7). TC Lionrock (2010) passed by a mooring of SCSMA in the SCS with the maximum wind speed of 24 m s$^{-1}$. The mooring was located 120 km to the right of the Lionrock track (Fig. S7a). TC Bill (2003) passed by the buoy 42041 of the NDBC in the NA with the same maximum wind speed of 24 m s$^{-1}$ as that in the SCS. Buoy 42041 was located ~110 km to the right of the Bill track (Fig. S7b). Therefore, the dynamical response comparison between the two TC cases is carried out under the same TC forcing conditions.

The raw time series of hourly horizontal current velocity are 48 h (2-day) high-pass filtered with a fourth-order Butterworth filter to extract internal wave currents as shown in Fig. 4a, c. The observed currents from the day of TC passage to about 20 days afterward are used to calculate the spectrum of horizontal current velocity and vertical shear of horizontal currents in Figs. 4b and S7c. Due to the lack of in situ observations of continuous temperature-salinity profiles under the two TCs, the climatological temperature-salinity profiles (Fig. S7e, f) from WOA13[57] in the corresponding months of TCs are used to calculate the inverse Richardson number ($Ri^{-1} = \frac{\frac{\partial u^2}{\partial z} + \frac{\partial v^2}{\partial z}}{\frac{g}{\rho_0} \frac{\delta \rho}{\delta z}}$) (Fig. S7d). A larger $Ri^{-1}$ indicates that the internal wave is more unstable, more liable to break and generate more intense turbulent entrainment[41,72].

## Nonlinear triad wave-wave interactions among internal waves

Nonlinear triad wave-wave interaction is an efficient way of cascading energy along the internal wave continuum from large-scale to small-scale internal waves (namely secondary waves), which commonly have smaller vertical wavelengths and are more easily dissipated[73]. Nonlinear couplings between near-inertial internal waves ($f$) and diurnal (D1) or semidiurnal (D2) ITs in the SCS have been well reported[74,75]. The frequencies of the generated secondary waves (termed $fD1$, $fD2$ and

$D2f$) are located at their sum or difference frequencies:

$$fD1 = f + D1 \quad (3)$$

$$fD2 = f + D2 \quad (4)$$

$$D2f = D2 - f \quad (5)$$

Reference 28 has reported that the nonlinear coupling between the TC-generated near-inertial internal waves and background diurnal ITs could transfer near-inertial energy to high-mode secondary wave $fD1$ and quickly dissipate the internal wave energy, triggering enhanced turbulent mixing locally. Further SCSMA and other mooring observations under more TC cases also support a nonlinear coupling between the TC-generated near-inertial internal waves and diurnal and semidiurnal ITs and the generation of secondary waves ($fD1$, $fD2$, $D2f$) (ref. 29). Generally, due to a much smaller vertical wavelength, these small-scale secondary waves could further enhance the vertical entrainment under the SCS TCs.

## The Price model experiments

In the present study, the TC-induced mixing scheme proposed by Price[42], hereafter the Price model, is used to evaluate impact of background ITs on TC-induced SST cooling. The Price model simulates similar SST cooling as the widely used three-dimensional Price-Weller-Pinkel model[76], but is computationally more efficient. It is a rationalized ocean metric for estimating SST after TC (named $T_d$) and hence cooling (pre-TC SST minus $T_d$) during a TC. The $T_d$ is estimated by averaging the initial (pre-TC) ocean temperature ($T(z)$) vertically:

$$T_d = \frac{1}{d} \int_{-d}^{0} T(z) dz \quad (6)$$

where $d$ is the depth of vertical mixing caused by a TC. According to ref. 42, the mixing depth could be estimated by Eq. (7):

$$\frac{g \delta \rho d}{\rho_0 (\frac{\tau}{\rho_0 d} \frac{4 R_h}{U_h} S)^2} \geq 0.65 \quad (7)$$

where $g$ is the acceleration of gravity. The $\rho_0$ is water density, which is taken as 1024 kg m$^{-3}$. $R_h$ and $U_h$ are the radius of maximum wind and TC translation speed, respectively. $\delta \rho$ is the density difference between the density at depth $d$ and that averaged from $d$ to the sea surface. $S$ is an empirical constant which accounts for the rotation effect. $\tau$ is the wind stress induced by the TC via:

$$\vec{\tau} = \rho_a C_D \vec{U}_{10} |U_{10}| \quad (8)$$

where $\rho_a$ is the density of air taken as 1.2 kg m$^{-3}$; $C_D$ is the drag coefficient based on ref. 77 to account for high-wind (>25 m s$^{-1}$) conditions; and $U_{10}$ is TC wind speed. The term $\frac{\tau}{\rho_0 d} \frac{4 R_h}{U_h} S$ represents the vertical shear at the base of the mixed layer (i.e., at depth $d$). In the present study, a total of 661 TC track points (6 hourly) in the SCS during 2000–2016 are used for the Price model experiments and the simulated SST cooling was compared with satellite observations. The initial temperature profile input to Price model experiments comes from the WOA13 monthly ocean temperature profile averaged in the TC season (Fig. S5). Salinity is kept constant at 33 psu.

Considering the strong background ITs in the SCS, we rewrite Eq. (7) and add a typical ITs' vertical shear of horizontal currents from

mooring observations ($\delta U_{IT}$) to account for the effects of ITs:

$$\frac{g\delta\rho d}{\rho_0(\frac{\tau}{\rho_0 d}\frac{4R_h}{U_h}S + \delta U_{IT})^2} \geq 0.65 \qquad (9)$$

Figure S9 shows the typical current velocity profile of ITs in the SCS used in the model experiments.

## The ROMS-WRF model experiments

The ROMS (Regional Ocean Modeling System) model was developed by Rutgers University and the University of California at Los Angeles; it solves the incompressible and hydrostatic primitive equations with a terrain-following vertical coordinate and is widely used in simulating upper ocean response to TCs[43,73,78]. In this study, the model domain, from 99°E to 135°E and from 10°S to 30°N, covers the entire SCS as shown by the black box in Fig. S10a. The model has a uniform horizontal resolution of 9 km and 40 layers in the vertical. The vertical mixing scheme applied in the simulations is the Mellor-Yamada scheme[79]. The topography is from a smoothed ETOPO2 to remove abrupt changes in topography to keep an accurate S-coordinate pressure gradient. The initial and lateral boundary conditions are from the Simple Ocean Data Assimilation (SODA)[80]. The surface momentum and heat fluxes are calculated from Bulk formulae[81]. The air temperature and humidity are kept unchanged as in National Centers for Environmental Prediction (NCEP) reanalysis data[82]. The 10 m surface wind is a synthetic TC wind field[83], combining the NCEP reanalysis data[82] and the JTWC best track.

We conduct two simulations with and without background ITs focusing on the TC Megi in October 2010 (Fig. S10). The without-ITs experiment is only forced by the TC winds (i.e., omitting tidal processes). To generate background ITs, the with-ITs experiment is forced by TC winds and also by four primary tidal constituents (K1, O1, M1 and S2) in the SCS, applied on the lateral open boundary. The four primary tidal constituents (K1, O1, M1 and S2) are from predictions of a barotropic tidal model TPXO7.2[84], which has been validated against observations in previous studies[27,85]. To quantify the effect of background ITs on SST cooling under TC Megi (2010), the difference of the mixing strength and TC-induced SST cooling between the two simulations (with-ITs minus without-ITs) is estimated and averaged along the 21 °N section as shown by the red line in Fig. S10a (120 km either side of the TC center).

The atmospheric model applied in this study is WRF (Weather Research and Forecasting) model. The Advanced Research dynamical core of WRF (ARW; version 3.8.1 here), is a state-of-art atmospheric simulation system designed for a broad range of meteorological and weather applications, which has been widely used in TC simulations[43]. Our model domain extends from 1°N to 28°N, and 104°E to 130°E, with grid resolution of 9 km in the horizontal and 35 vertical levels. The physics options used in the model are shown in Table S2.

To quantify the sensitivity of TC intensity to the ITs-heightened SST cooling, we conducted two idealized WRF ensemble simulations focusing on the abovementioned TC Megi (2010). Our sensitivity experiment strategy was similar to the two-tiered approach used in ref. 43. The simulated SST in ocean model (ROMS) is used as the surface boundary condition over the ocean to force the atmospheric WRF model. In the without-ITs and with-ITs experiments, the surface boundary condition over the ocean, i.e., the SST, was obtained from the real-time output of the without-ITs and with-ITs ROMS simulation, respectively. Except for SST, all other forces and factors were identical for both ensemble experiments, based on data from the NCEP-Climate Forecast System Reanalysis (CFSR). Each ensemble experiment comprises four simulations, having initial conditions generated by the WRF model's built-in method RANDOMCV. All the simulations were initialized on 0000 UTC 18 October and ended on 0000 UTC 23 October 2010, since these reproduced TC tracks were most closely to JTWC.

The corresponding 4 tracks and mean tracks are shown in Fig. S10b. TC Megi's intensity sensitivity to SST in our WRF simulations is shown in Figs. S11 and S12. Overall, ROMS-WRF sensitivity experiments demonstrated suppression related to background ITs on SCS TC intensification via heightening of the TC-ocean coupled cooling effect.

## Data availability

The TC best track data are obtained from the IBTrACS TC dataset (https://climatedataguide.ucar.edu/climate-data/ibtracs-tropical-cyclone-best-track-data). The WOA13 dataset is available at https://www.nodc.noaa.gov/OC5/woa13/. SST data are obtained from the RSS (http://www.remss.com/). The data used to calculate potential intensity and vertical wind shear are from the ECMWF's Interim Reanalysis database (http://apps.ecmwf.int/datasets/). The buoy data are available from the NDBC (http://www.ndbc.noaa.gov/). The SCSMA mooring datasets used in this study are available from the corresponding authors upon request. The data used for plotting the figures in the paper are available from https://doi.org/10.5281/zenodo.10878392. Source data are provided with this paper.

## Code availability

The codes used for generating the figures in the paper can be accessed at https://doi.org/10.5281/zenodo.10878392.

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

## Acknowledgements

This study was supported by the National Key Research and Development Program of China (2022YFC3104304 of S.G.), the National Natural Science Foundation of China (92258301 of J.T., and 41876011 of S.G.), and the 2019 Research Program of Sanya Yazhou Bay Science and Technology City (SKJC-KJ-2019KY04 of W.Z.). F.J. is supported by the U.S. National Science Foundation (AGS-1813611) and the U.S. Department of Energy (DE-SC0005110). I.I.L. is supported by the Taiwan's Ministry of Science and Technology (103-2111-M-002-002-MY3).

## Author contributions

J.T., F.F.J., I.I.L. and S.G. conceived the central idea. S.G. and I.F.P. performed most of the analyses and generated the figures. I.I.L., W.C., F.F.J. and S.G. wrote the main manuscript with further inputs from J.T. and W.Z. J.T. and W.Z. designed and conducted observations of the South China Sea Mooring Array, collected the mooring data and participated in analyzing the data with S.G. Z.X. performed the ROMS and WRF model experiments and analyzed the model outputs. J.H., Z.J., L.Z., P.L., Y.Z., Z.Z., C.Z., Q.Y., X.H., Y.H. and J.S. contributed to the discussion of the results and commented on the manuscript.

## Competing interests

The authors declare no competing interests.
