## [Peer Review File · Nature Communications]

Ocean Internal Tides Suppress Tropical Cyclones in the South China SeaEditorial Note: Parts of this Peer Review File have been redacted as indicated to maintain the confidentiality of unpublished data.

REVIEWER COMMENTS

Reviewer #1 (Remarks to the Author):

Very interesting paper. Well written, and a pleasure to read. Worthy of publication in a major, international journal.

My only comment is minor and stylistic. I suggest that there should be no reference to an implied purpose to anything that happens in the ocean/atmosphere. For example, the title could read 'Ocean internal tides suppress tropical cyclones ...'. The phrase that I suggest be omitted 'as an effective', implies that there is a preferred outcome (there is, but it is irrelevant to this analysis) or that the ocean has a purpose (most definitely not). Once you recognize this stylistic foible, you will be able to expunge the multiple occurrences easily.

Reviewer #2 (Remarks to the Author):

Review of "Ocean internal tides as an effective suppressor of tropical cyclones in the South China Sea"

Dear Editor,

This submission lays out statistics showing how TC intensification is exceedingly rare in the South China Sea. It aims to discard multiple explanations and lastly argues that excess mixing produced by interactions between internal tides and TC-generated near-inertial internal waves (NIWs) are responsible for the lack of intensification in the South China Sea.

Even though the authors present some compelling and intriguing arguments, I believe this work to be very far from ready for publication. Most importantly, the work needs thorough editing and curation, as well as a quality introduction. In its current form, the submission dedicates too many words and figures to ideas and processes that are besides the main point. It runs through all the analyses performed to reach a result, so the stronger arguments are lost in the mix.

I recommend that this submission be rejected. The authors may have a story worth telling, but they need to discern what information is essential to the story and do a better job focusing on those crucial pieces of information. Also, the present submission does not have an introduction and jumps into results from the very first paragraph.

Major comments:

1. The authors need to write a proper introduction. The current manuscript jumps in to deliver results

without having set the stage for the topic and its relevance, or formulated a research question. As it's written now, this introduction is overwhelming and confusing. The statistical analyses of TC tracks do a good job laying out the problem, comparing the South China Sea to other basins and pointing out especially how the local TC statistics stand out. However, the text doesn't do the analyses justice and the authors need to make sure that they have stated the problem clearly before they start trying to answer it. There also needs to be a more thorough discussion of past work relating to TC activity in the SCS from a climate/atmospheric perspective. What do atmospheric scientists say drives variability in TC tracks in the SCS?

2. Lines 111-121: The discussion of mechanisms controlling SST cooling under TCs here is too reductive. It is true that SST cooling can scale with stratification, TC intensity, and translation speed. However, these are not general rules as a long list of exceptions applies. The authors are hinting at the idea that ITs make SST cooling greater in the SCS than elsewhere, but the statistical tests in Fig. 2 don't directly address that question. In particular, I don't think that Figs 2C and 2D are particularly meaningful to the story being told.

3. In-situ observations: The potential for near-inertial wave generation varies greatly across different locations relative to the TC center. Moreover, this changes drastically depending on TC characteristics and the background quasigeostrophic environment. Therefore, it's difficult to make any general conclusions from the comparison of measurements presented here. I don't think this comparison is useful, or that it adds anything to the authors' case.

4. Spectral analyses: Motions at frequencies $fD1$, $fD2$ most often result from contamination rather than nonlinear effects. To discard contamination, the authors ought to interpolate observed velocities onto isopycnals and then perform spectral analysis on the time series of velocity along individual isopycnals, rather than fixed depth levels. For a more detailed explanation of this contamination, see Sherman and Pinkel 1991 and Le Boyer et al. 2020.

5. The most compelling argument pointing to internal tides as the culprits for low intensification rates, in my opinion, comes from the modeling experiments with and without tides. However, the presentation of modeling results makes it difficult to distinguish between the background effects of including tides, and the effects of tides interacting with TC motions. For example, was the SST already colder in the Tide simulation even before the storm? Figure 4D suggests that it was.

Reviewer #3 (Remarks to the Author):

In this manuscript, the authors ascribed the weak intensification of TCs over the SCS to the Moon-induced internal tides. I think this is a novel viewpoint to understand the TC activity in this region, and the conclusions will help improve the numerical simulation and forecast of the TCs. I have some minor comments on this version before considering acceptance of publication.

1. The authors should discuss the influences of monsoon flow. The basic flow around the SCS is different between June and October because of the seasonal cycle of monsoon. Since the Moon-induced ITs are stable (Fig. S7), the energy flux may change with the monsoon circulation's seasonal variation.
2. Does the TC-IT interaction depend on the TC track? The IT effects intensify the oceanic current evidently; what about the contribution of variation in mixed layer depth and barrier layer thickness?
3. What about the islands' effects on the weak TC intensification over the SCS? The islands may increase the surface roughness and decrease the TC intensification.
4. Have you checked more cases to ascertain the differences in TCs between SCS and North Atlantic? Why not compare the TCs in the SCS and northern IO since they have more similar monsoon climate backgrounds?
5. How to understand the stronger oceanic current but weaker TC-related winds? Because the entrainment cooling is sensitive to the surface winds.

Ref. Number: NCOMMS-23-37157-T
Title: Ocean Internal Tides as An Effective Suppressor of Tropical Cyclones in the South China
Sea (Now revised to ‘Ocean Internal Tides Suppress Tropical Cyclones in the South China
Sea’)

**Response to Reviewers**

We would like to thank the reviewers for their constructive comments and suggestions.
We have addressed the reviewers’ concerns and implemented useful suggestions in our revised
manuscript. The point-to-point responses are listed below in **blue**.

**Responses to Reviewer #1**

Very interesting paper. Well written, and a pleasure to read. Worthy of publication in a major,
international journal.

Dear Jim,

Thank you so much for the positive evaluation of our manuscript and we are grateful for your
encouragement. We have revised the manuscript by carefully addressing your comments as
follows.

My only comment is minor and stylistic. I suggest that there should be no reference to an
implied purpose to anything that happens in the ocean/atmosphere. For example, the title could
read 'Ocean internal tides suppress tropical cyclones ...'. The phrase that I suggest be omitted
'as an effective', implies that there is a preferred outcome (there is, but it is irrelevant to this
analysis) or that the ocean has a purpose (most definitely not). Once you recognize this stylistic
foible, you will be able to expunge the multiple occurrences easily.

Yours, Jim Price.

Thank you very much for this valuable comment. We agree with you that using the phrase “as
an effective” is a stylistic foible, as oceanic internal tide itself definitely does not have a purpose
to suppress tropical cyclones. As you suggested, the title has been modified to “Oceanic
Internal Tides Suppress Tropical Cyclones in the South China Sea”. This stylistic foible
occurring in other parts has also been corrected.

Responses to Reviewer #2

Review of “Ocean internal tides as an effective suppressor of tropical cyclones in the South
China Sea”

This submission lays out statistics showing how TC intensification is exceedingly rare in the
South China Sea. It aims to discard multiple explanations and lastly argues that excess mixing
produced by interactions between internal tides and TC-generated near-inertial internal waves
(NIWs) are responsible for the lack of intensification in the South China Sea.

Even though the authors present some compelling and intriguing arguments, I believe this work
to be very far from ready for publication. Most importantly, the work needs thorough editing
and curation, as well as a quality introduction. In its current form, the submission dedicates too
many words and figures to ideas and processes that are besides the main point. It runs through
all the analyses performed to reach a result, so the stronger arguments are lost in the mix.

I recommend that this submission be rejected. The authors may have a story worth telling, but
they need to discern what information is essential to the story and do a better job focusing on
those crucial pieces of information. Also, the present submission does not have an introduction
and jumps into results from the very first paragraph.

We are grateful for your thorough review, comments and suggestions. Thank you for your
recognition of our key arguments. We understand your concerns and have substantially revised
the manuscript by carefully considering and addressing your comments and suggestions.
Specifically, the revised version is more focused, has an improved introduction, and has moved
the result to the main section after the introduction.

The logic flow for Figs 1–5 in the revised manuscript is organized as follows.

(1) Opening and setting-up the scientific question of the manuscript, that is, SCS TCs have the
globally lowest intensification characteristics despite the high frequency.

(2) Examining the possible atmospheric and oceanic factors related to TC intensification and
points to the ocean cooling effect being the key factor among all the factors examined.

(3) Examining the possible factors influencing the TC-induced SST cooling and pointing out
none of the known factors (e.g., TC translation speed, size, and stratification) can explain the
unusually strong cooling effect in the SCS.

(4) Using in-situ observation to demonstrate that the presence of oceanic background internal
tides can heighten the cooling effect.

(5) Using numerical modelling experiments to illustrate that the presence of background
oceanic internal tides can enhance subsurface turbulent mixing and heighten the cooling effect.

Detailed point-to-point responses to your comments are listed below. If there are still places of
misunderstanding or missing of your points, please kindly let us know also.

Major comments:

1. The authors need to write a proper introduction. The current manuscript jumps in to deliver
results without having set the stage for the topic and its relevance, or formulated a research
question. As it's written now, this introduction is overwhelming and confusing. The statistical
analyses of TC tracks do a good job laying out the problem, comparing the South China Sea to
other basins and pointing out especially how the local TC statistics stand out. However, the
text doesn't do the analyses justice and the authors need to make sure that they have stated the
problem clearly before they start trying to answer it. There also needs to be a more thorough
discussion of past work relating to TC activity in the SCS from a climate/atmospheric
perspective. What do atmospheric scientists say drives variability in TC tracks in the SCS?

Thank you for the suggestion. We agree with you that the manuscript should have a proper
introduction without presenting results in the introduction. We have moved the detailed result,
originally in the introduction, to the result section. We have made a substantial revision to the
introduction. We first review relevant atmospheric and oceanic factors modulating TC
intensification reported in the literature (e.g., Bister & Emanuel 1998; Frank & Ritchie 2001;
Lin & Chan 2015). We then detail the negative feedback of TC self-induced SST cooling to
TC intensification (Chang & Anthes 1979; Price 1981; Emanuel, 1999; Bender & Ginis 2000;
D'Asaro et al. 2007). In particular, we highlight that the TC self-induced SST cooling
suppresses TC intensification based on both model experiments and observations (Bender &
Ginis 2000; D'Asaro et al. 2007; Lloyd & Vecchi 2011, etc.). This is followed by a review of
the modulating effect of the pre-TC presence of multiple oceanic motions or features, such as
mesoscale eddies (Shay et al. 2000; Lin et al. 2005) or barrier layers (Balaguru et al. 2012), as
well as the potential effects of internal tides (ITs) in this study. We then briefly introduce some
unique TC characteristics in the South China Sea (SCS), focusing on its globally lowest
intensification characteristics. The associated dynamics, which remain unknown, are then
raised. We end the introduction by using a sentence on our findings as per the format of the
journal.

From the climate/atmospheric perspective, previous studies on SCS TCs mainly focused on
their genesis and interannual to decadal variability, and found that they are related to the East
Asia monsoon, ENSO, Pacific Decadal Oscillation, and so on (e.g., Wang et al. 2007; Zuki &

Lupo 2008; Goh & Chan 2010; Lee et al. 2012). To date, few studies examine the weak TC
intensification in the SCS. This is briefly reviewed in the Introduction (Lines 62–67 in the
revised clean version).

Previously, atmospheric scientists may instinctually attribute the scarcity of intense TCs in the
SCS to the small ocean basin and landmass (or islands) effects (Brand & Blelloch 1974; Chou
et al. 2011). In this study, we carefully examine the atmospheric factors and the effects from
basin size and landmass (or islands), and find that these effects are not responsible for the
weakest TC intensification in the SCS. We describe these in the result section (Lines 104–120
in the revised clean version).

2. Lines 111-121: The discussion of mechanisms controlling SST cooling under TCs here is
too reductive. It is true that SST cooling can scale with stratification, TC intensity, and
translation speed. However, these are not general rules as a long list of exceptions applies. The
authors are hinting at the idea that ITs make SST cooling greater in the SCS than elsewhere,
but the statistical tests in Fig. 2 don't directly address that question. In particular, I don't think
that Figs 2C and 2D are particularly meaningful to the story being told.

As pointed out by the reviewer and many previous studies (e.g., Price 1981; Lloyd & Vecchi
2011; Vincent et al. 2012; Mei & Pasquero 2013; D'Asaro et al. 2014), TC-induced SST
cooling is mainly influenced by TC characteristics (including TC intensity, translation speed,
and storm size), as well as by pre-TC upper ocean thermal stratification (Mei & Pasquero 2013;
Liu et al. 2023). Potential exceptions that are not discussed in the study are the effect of pre-
TC existing mesoscale eddies and barrier layer, which could also modulate the magnitudes of
TC-induced SST cooling by altering the upper ocean thermal structure (Jaimes & Shay 2009;
Balaguru et al. 2012). Specifically, the pre-TC presence of warm (cold) eddies tends to reduce
(enlarge) the cooling (Jaimes & Shay 2009). The barrier layer can suppress upper ocean mixing
during TC passage and thus reduce the cooling (Balaguru et al. 2012). The aforementioned
factors encompass nearly all the known factors influencing TC-induced SST cooling.

Through global comparisons, we find that neither the TC characteristics nor the upper ocean
thermal structure can explain the strongest cooling induced by TCs in the SCS. In addition,
mesoscale eddies or barrier layers, not discussed in the original manuscript, cannot explain this
strong cooling in the SCS either. Both mesoscale warm and cold eddies are highly active in the
SCS (Wang 2003), and their encounters with TCs do not exhibit a preference for either polarity;

correspondingly, their modulating effect on cooling should have no preference. During summer
and early autumn when TCs are most active, the SCS is characterized by a thick and frequent
barrier layer (Zeng & Wang 2017) and certainly cannot be responsible for the strong cooling.

Our **Fig. 2** aims to rule out the possibility that known factors result in the unusually strong SST
cooling in the SCS. In-situ observations in **Fig. 3** and modelling experiments in **Fig. 4** address
this issue by presenting evidence that background ITs enhance turbulent mixing and enlarge
TC-induced SST cooling.

In the original manuscript, **Figs 2C** and **2D** are used to elucidate that the ocean thermal
stratification may not be the cause for the strongest cooling in the SCS via a comparison with
the Eastern Pacific. We agree with you that **Figs 2C** and **2D** seem not particularly meaningful
to the whole story and we have moved **Figs 2C** and **2D** to supplementary figures. Following
your suggestions to focus more on the main point, we substantially revised **Fig. 2** by
emphasizing and comparing environmental factors influencing TC intensification between
SCS and other TC-active oceans. To strengthen the analysis of SST cooling, we added a new
figure as **Fig. 3** to elucidate that neither TC attributes nor ocean stratification is the cause for
the strongest SST cooling in the SCS. Then **Fig. 4** (original **Fig. 3**) based on in-situ observations
and **Fig. 5** (original **Fig. 4**) based on modelling experiments demonstrate that background ITs
enhance turbulent mixing and enlarge TC-induced SST cooling.

3. In-situ observations: The potential for near-inertial wave generation varies greatly across
different locations relative to the TC center. Moreover, this changes drastically depending on
TC characteristics and the background quasigeostrophic environment. Therefore, it's difficult
to make any general conclusions from the comparison of measurements presented here. I don't
think this comparison is useful, or that it adds anything to the authors' case.

Thank you for the comments. We agree with you that the generation of near-inertial waves
(NIWs) depends on multiple factors. We have taken these factors into consideration when
searching for comparable cases. In the original manuscript, the comparison of upper ocean
currents is under similar TC conditions, such as relative locations to TC center, TC intensity,
and translation speed. Following your suggestions, we further compare the quasigeostrophic
environment, shown by the sea level anomaly (SLA) and geostrophic current before the
passages of both SCS and North Atlantic (NA) TCs (**Fig. R1**). Results show that both the
mooring and the buoy are located out of eddy regimes, with weak geostrophic currents.

Therefore, the quasigeostrophic environment should have a negligible impact on the observed
currents. In summary, the upper ocean responses to SCS TC and NA TC in this study are
compared under similar conditions. Therefore, we think that the comparison is useful and
illustrates the effect of background ITs.

**Fig. R1 Sea level anomaly and geostrophic current before the passage of TCs over SCS and North**
**Atlantic.** **a** Sea level anomaly (color) and geostrophic current (black arrow) fields before TC Lionrock
(2010) passing the mooring (red star) in the SCS. **b** Sea level anomaly (color) and geostrophic current
(black arrow) fields before TC Bill (2003) passing the buoy (blue star) in the North Atlantic. Black solid
lines and dots denote the tracks of TC Lionrock and Bill. The sea level anomaly and geostrophic current
data are obtained from the Copernicus Marine Environment Monitoring Service.

We agree with you that it is difficult to make a general conclusion based on only one case for
comparison. Therefore, we search in-situ observations from global mooring/buoy databases
and compare using more TC cases; the results based on these cases are consistent with those
shown in the original manuscript. The mooring/buoy observations for comparison should have
upper ocean current observations and should be subject to similar TC characteristics. Based on
the South China Sea Mooring Array (SCSMA) constructed by the Ocean University of China
since 2009, we finally obtain hundreds of cases where moorings are located within 200 km
from the TC center (**Figs R2a** and **R2c**). In other TC-active ocean basins, available mooring or
buoy observations with subsurface current information during TC passages are very rare. To
the best of our ability, we obtain three buoys (42002, 42038, and 42041) with upper ocean
current observations in the NA from the National Data Buoy Center. The corresponding NA
TC tracks and buoys are shown in **Figs R2b** and **R2d**. In total, 14 cases in which buoys recorded
the upper ocean current response to NA TCs are identified.

Based on the TC-mooring/buoy observational cases shown in **Fig. R2**, more than 10 pairs of
comparable cases are identified (**Table R1**). For simplicity, four pairs are shown here (**Figs**
**R3–R6**). Consistent with results in **Fig. 4**, **Figs R3–R6** show that upper ocean responses to
SCS TCs manifest more complicated and stronger currents and shears than the responses to
NA TCs. Power spectra of upper ocean currents during SCS TCs show energetic NIWs
generated by TCs and also background ITs ($D1$, $D2$). In addition, energetic spectral peaks
appear at frequencies of secondary waves ($fD2$, $fD2$, $D2f$), indicating the nonlinear coupling
between TC-generated NIWs and background ITs, evidenced also in the power spectra of
along-isopycnal currents (for details please refer to the response to Comment 4). However, the
power spectra of the upper ocean currents during NA TCs only show energetic peaks of TC-
generated NIWs. On average, the integrated internal wave energy during SCS TCs is more than
five times larger than that during NA TCs, which can effectively enhance the upper ocean
mixing and enlarge SST cooling.

Overall, based on observations from these new cases, we think that the interactions between
background ITs and TC-generated NIWs likely amplify turbulent entrainment and thus enlarge
the SST cooling.

[REDACTED]

**Fig. R2.** [REDACTED]

**Table R1.** TC and mooring/buoy information in the comparable TC cases between the South China Sea
 (SCS) and the North Atlantic (NA).

Ocean basin	TC name	Wind speed (m s ⁻¹)	Translation speed (m s ⁻¹)	Distance (km)
SCS	Kompasu (2021)	28.3	6.8	222
NA	Dolly (2008)	30.9	4.9	231
SCS	Haitang (2017)	20.6	7.6	-145
NA	Ivan (2004)	18.0	7.7	-144
SCS	Atsani (2020)	23.1	4.4	-65
NA	Matthew (2004)	20.6	4.1	-88
SCS	Hato (2017)	25.7	6.6	127
NA	Bill (2003)	23.1	6.0	102
SCS	Cimaron (2013)	20.6	8.7	142
NA	Erika (2003)	23.1	9.8	156
SCS	Thirteen (2018)	18.0	6.0	-144
NA	Ivan (2004)	18.0	7.7	-144
SCS	Danas (2019)	18.0	7.8	-144
NA	Ivan (2004)	18.0	7.7	-144
SCS	Barijat (2018)	20.6	4.4	-90
NA	Matthew (2004)	20.6	4.1	-88
SCS	Choi-wan (2021)	18.0	4.8	-88
NA	Matthew (2004)	20.6	4.1	-88
SCS	Barijat (2018)	20.6	4.4	95
NA	Bill (2003)	23.1	6.0	102
SCS	Mekkhala (2020)	23.1	5.9	102
NA	Bill (2003)	23.1	6.0	102

* ‘Wind speed’ and ‘translation speed’ mean the maximum wind speed and translation speed of the TC
 when TC passes the mooring/buoy. ‘Distance’ represents the distance between the TC center and
 mooring/buoy, and positive (negative) sign means the mooring/buoy is located to the right (left) of the
 TC track.

 **Fig. R3. Comparisons of upper ocean response to SCS TC (with ITs) and NA TC (without ITs). a**
 **TC track and mooring position (red star) in the SCS. b TC track and buoy position (blue star) in the NA.**
 **The time when TC passed mooring (buoy) is written in the bottom left corner of (a) and (b). c Mooring**
 **observed ocean horizontal currents during SCS TC passage. d Buoy-observed ocean horizontal currents**
 **during NA TC passage. The currents in (c) and (d) are 48 h high-pass filtered with a fourth-order**
 **Butterworth filter. e Power spectrum density (PSD) of horizontal currents along fixed depths during**
 **and after the passage of SCS TC (red) and NA TC (blue). The PSD is averaged over the upper 150 m.**
 **f Power spectrum density (PSD) of horizontal currents along isopycnals during and after the passage of**
 **SCS TC (red) and NA TC (blue). The PSD in the SCS is averaged over isopycnals marked by dark lines**
 **in (c). Due to the lack of temperature observations, the PSD in the NA is still averaged over the upper**
 **150 m.**

**Fig. R4.** Same as **Fig. R3**, but for a different pair of TC cases.

**Fig. R5.** Same as **Fig. R3**, but for a different pair of TC cases.

**Fig. R6.** Same as **Fig. R3**, but for a different pair of TC cases.

4. Spectral analyses: Motions at frequencies $fD1$, $fD2$ most often result from contamination
 rather than nonlinear effects. To discard contamination, the authors ought to interpolate
 observed velocities onto isopycnals and then perform spectral analysis on the time series of
 velocity along individual isopycnals, rather than fixed depth levels. For a more detailed
 explanation of this contamination, see Sherman and Pinkel 1991 and Le Boyer et al. 2020.

We agree with you that spectral peaks at frequencies $fD1$, $fD2$ might result from fine-structure
 contamination under Eulerian coordinate due to the Doppler shifting effects of internal tides.
 Sherman and Pinkel (1991) proposed a useful method to reduce this contamination by
 interpolating velocities onto a semi-Lagrangian or isopycnal-following frame, which works
 well for reducing the fine-structure contamination effect as shown in Le Boyer et al. (2020).

Following the method proposed in Sherman and Pinkel (1991) and Le Boyer et al. (2020), here
 we utilize the current velocities along individual isopycnals to calculate the power spectra.
 Mooring observations under the TC cases in Fig. 4 lack temperature/salinity (T/S) data in the
 upper ocean, limiting us to interpolate current velocity onto isopycnals. After an extensive
 search of the TC cases in **Figs R2a** and **R2c**, we select a total of 35 TC cases captured by
 moorings with synchronously subsurface current and T/S observations. We interpolate the

current velocities observed at fixed depth levels onto isopycnals, and then perform a spectral
analysis using current velocities along individual isopycnals. Finally, the power spectra are
averaged across different isopycnals. As an example, **Fig. R7** shows the mooring-observed
current velocity and corresponding spectra under one TC case. The mooring is located 59 km
right to the TC track (**Fig. R7a**). When the TC passes the mooring, its maximum wind speed
and translation speed are 30.9 m s^{-1} and 6.4 m s^{-1} , respectively.

Filtered current velocities reveal the strengthening of currents at NIW and secondary wave
frequencies during TC passage, both for currents at fixed depth levels and along individual
isopycnals (**Figs R7c–R7f**). As expected, power spectra show energetic peaks at near-inertial
(f), diurnal (D1), and semidiurnal (D2) frequencies. Additionally, evident spectral peaks also
appear at frequencies of secondary waves ($fD1$, $fD2$, $D2f$), indicating the nonlinear coupling
between TC-generated NIWs and background ITs. Power spectra of currents along isopycnals
under more TC cases, showing the nonlinear coupling effect (**Figs R3f–R6f**). Furthermore, we
averaged power spectra under 35 TC cases to generate a composite power spectrum (**Fig. R8**),
which also shows spectral peaks at frequencies of secondary waves. Power spectra in all these
cases, calculated by along-isopycnal current to discard the fine-structure contamination, give
observational evidence for the nonlinear coupling between background ITs and TC-generated
NIWs.

**Fig. R7. Ocean response to SCS TC.** **a** TC track and mooring position (red star) in the SCS. The time
 when TC passed mooring is indicated in the bottom left corner. **b** Mooring-observed density during TC
 passage. Black solid lines denote isopycnals along which spectral analysis is performed. **c** 48 h high-
 pass filtered ocean horizontal currents with a fourth-order Butterworth filter. **d** Same as (c) but for
 horizontal currents along isopycnals. **e** Band-pass filtered ocean horizontal currents with the band
 $[0.9(f+Q1) \ 1.1(f+K1)]$. **f** Same as (e), but for horizontal currents along isopycnals. **g** Power spectrum
 density (PSD) of horizontal currents during and after the TC passage. The PSD is averaged over the
 upper 150 m. **h** Power spectrum density (PSD) of horizontal currents along isopycnals during and after
 the TC passage. The PSD is averaged across the isopycnals indicated by black solid lines in (b).

**Fig. R8. Composite power spectrum density (PSD) of ocean horizontal currents during and after**
 **the passage of SCS TCs. a** PSD of ocean currents over the upper 150 m. **b** The corresponding PSD of
 ocean currents along isopycnals.

5. The most compelling argument pointing to internal tides as the culprits for low
 intensification rates, in my opinion, comes from the modeling experiments with and without
 tides. However, the presentation of modeling results makes it difficult to distinguish between
 the background effects of including tides, and the effects of tides interacting with TC motions.
 For example, was the SST already colder in the Tide simulation even before the storm? Figure
 4D suggests that it was.

Thank you for your acknowledgment of our modeling experiments. We think there may be
 some misunderstanding on the **Fig. 4d** in the original manuscript (**Fig. 5d** in the revised
 manuscript). In this figure, “0 hour relative to TC” represented by the black dashed line
 indicates the exact time when the TC center reaches the target section. However, the high winds
 associated with TCs usually cover hundreds of kilometers from the TC center. Thus, the front
 edge of TC high winds usually arrives at target section tens of hours earlier than the TC center.
 For TC Megi (2010) here, the radius of 64 kt wind speeds (i.e., typhoon-force wind) is
 approximately 150 km in the SCS (Pun et al. 2018). Therefore, the front edge of Megi has
 begun influencing the analyzed target section roughly 12–14 hours prior to the arrival of the
 TC center (**Fig. R9**). Therefore, the colder SST before storm center passage (0 h) in the with-
 ITs simulation is not due to background effects of including tides, but due to the ITs-NIW's
 interaction under the influence of TC's front part.

**Fig. R9. Schematic diagram of the area influenced by a TC.** Black vertical solid line denotes the
 analyzed target section, i.e., the section in Supplementary Fig. S11a. Red circular line represents the 64
 317 kt winds of TC 12 hour before the TC center passing the target section (-12 h, -150 km). Red arrow
 indicates the moving direction of the TC. Black circular line represents the 64 kt winds of TC at the
 exact time of the TC center just over the target section (0 h, 0 km).

**[REDACTED]**

[REDACTED]

**Fig. R10.** [REDACTED]

References:

Balaguru, K. et al. Ocean barrier layers' effect on tropical cyclone intensification. *Proc. Natl.*
*Acad. Sci. U.S.A.* **109** (36) 14343-14347 (2012).

Bender, M. A. & Ginis, I. Real-case simulations of hurricane–ocean interaction using a high-
resolution coupled model: Effects on hurricane intensity. *Mon. Wea. Rev.* **128**(4), 917-946
(2000).

Bister, M. & Emanuel, K. A. Dissipative heating and hurricane intensity. *Meteor. Atmos. Phys.*
**52**, 233-240 (1998).

Brand, S. & Blelloch, J. W. Changes in the characteristics of typhoons crossing the island of
Taiwan. *Mon. Wea. Rev.* **102**(10), 708-713 (1974).

Chang, S. W. & Anthes, R. A. The mutual response of the tropical cyclone and the ocean. *J.*
*Phys. Oceanogr.* **9**(1), 128-135 (1979).

Chou, K. H., Wu, C. C. & Wang, Y. Eyewall Evolution of Typhoons Crossing the Philippines
and Taiwan: An Observational Study. *Terrestrial, Atmospheric & Oceanic Sciences* **22**(6)
(2011).

D'Asaro, E. A., Sanford, T. B., Niiler, P. P. & Terrill, E. J. Cold wake of hurricane Frances.
*Geophys. Res. Lett.* **34**(15) (2007).

D'Asaro, E. A. et al. Impact of typhoons on the ocean in the Pacific. *Bull. Amer. Meteor. Soc.* **95**,
1405-1418 (2014).

Emanuel, K. A. Thermodynamic control of hurricane intensity. *Nature* **401**(6754), 665-669
(1999).

Frank, W. M. & Ritchie, E. A. Effects of vertical wind shear on the intensity and structure of
numerically simulated hurricanes. *Mon. Wea. Rev.* **129**(9), 2249-2269 (2001).

Goh, A. Z. C. & Chan, J. C. Interannual and interdecadal variations of tropical cyclone activity
in the South China Sea. *International Journal of Climatology: A Journal of the Royal*
*Meteorological Society* **30**(6), 827-843 (2010).

Jaimes, B. & Shay, L. K. Mixed layer cooling in mesoscale oceanic eddies during Hurricanes
Katrina and Rita. *Mon. Wea. Rev.*, **137**(12), 4188-4207 (2009).

Le Boyer, A., Alford, M. H., Pinkel, R., Hennon, T. D., Yang, Y. J., Ko, D., & Nash, J.
Frequency shift of near-inertial waves in the South China Sea. *J. Phys. Oceanogr.* **50**(5), 1121-
1135 (2020).

Lee, T. C., Leung, C. Y. Y., Kok, M. H., & Chan, H. S. The long term variations of tropical
cyclone activity in the South China Sea and the vicinity of Hong Kong. *Tropical Cyclone*
*Research and Review* **1**(3), 277-292 (2012).

Lloyd, I. D. & Vecchi, G. A. Observational evidence for oceanic controls on hurricane intensity.
*J. Clim.* **24**, 1138-1153 (2011).

Lin, I. I. & Chan, J. C. Recent decrease in typhoon destructive potential and global warming
implications. *Nat. Commun.* **6**(1), 7182 (2015).

Lin, I. I., Wu, C. C., Emanuel, K. A., Lee, I. H., Wu, C. R. & Pun, I. F. The interaction of
Supertyphoon Maemi (2003) with a warm ocean eddy. *Mon. Wea. Rev.* **133**(9), 2635-2649
(2005).

Liu, Y. et al. Effect of storm size on sea surface cooling and tropical cyclone intensification in
the western north Pacific. *J. Clim.* **36**(20), 7277-7296 (2023).

Mei, W. & Pasquero, C. Spatial and temporal characterization of sea surface temperature
response to tropical cyclones. *J. Clim.* **26**(11), 3745-3765 (2013).

Price, J. F. Upper ocean response to a hurricane. *J. Phys. Oceanogr.* **11**, 153-175 (1981).

Pun, I. F., Lin, I. I., Lien, C. C. & Wu, C. C. Influence of the size of Supertyphoon Megi (2010)
on SST cooling. *Mon. Wea. Rev.* **146**(3), 661-677 (2018).

Shay, L. K., Goni, G. J. & Black, P. G. Effects of a warm oceanic feature on Hurricane Opal.
*Mon. Wea. Rev.* **128**(5), 1366-1383 (2000).

Sherman, J. T. & Pinkel, R. Estimates of the vertical wavenumber–frequency spectra of vertical
shear and strain. *J. Phys. Oceanogr.* **21**(2), 292-303 (1991).

Vincent, E. M. et al. Processes setting the characteristics of sea surface cooling induced by
tropical cyclones. *J. Geophys. Res.* **117**, C02020 (2012).

Wang, G. Mesoscale eddies in the South China Sea observed with altimeter data. *Geophys. Res.*
*Lett.* **30**(21) (2003).

Wang G., Su J., Ding Y. & Chen D. Tropical cyclones genesis over the South China Sea.
*Journal of Marine Systems* **68**(3-4), 318-326 (2007).

- Zeng, L. & Wang, D. Seasonal variations in the barrier layer in the South China Sea:
characteristics, mechanisms and impact of warming. *Clim. Dyn.* **48**, 1911–1930 (2017).
- Zuki, Z. M. & Lupo, A. R. Interannual variability of tropical cyclone activity in the southern
South China Sea. *J. Geophys. Res.* **113**(D6) (2008).

Responses to Reviewer #3

In this manuscript, the authors ascribed the weak intensification of TCs over the SCS to the
Moon-induced internal tides. I think this is a novel viewpoint to understand the TC activity in
this region, and the conclusions will help improve the numerical simulation and forecast of the
TCs. I have some minor comments on this version before considering acceptance of publication.
Thank you very much for the positive evaluation of our manuscript. We have revised the
manuscript by carefully addressing your comments and suggestions. Detailed point-to-point
responses to your comments are listed below.

1. The authors should discuss the influences of monsoon flow. The basic flow around the SCS
is different between June and October because of the seasonal cycle of monsoon. Since the
Moon-induced ITs are stable (Fig. S7), the energy flux may change with the monsoon
circulation's seasonal variation.

Thank you very much for the insightful comment. The basic atmospheric flow in the SCS
shows a seasonal cycle controlled by the East Asian Monsoon (Wang et al. 2009). Generally,
during May–September the southwesterly wind prevails (the summer monsoon), while during
October–April the northeasterly wind prevails (the winter monsoon). The seasonal monsoon
influences the ocean circulation and stratification in the SCS (Zu et al. 2019). Considering the
fact that the generation of internal tides (ITs) is highly associated with variations in the abyssal
stratification (Müller 2013), we agree with you that the monsoon flow may affect the seasonal
variation of internal tidal energy flux through influencing the ocean circulation and
stratification.

The topic on seasonal variation of internal tidal energy flux influenced by monsoon flow itself,
is a very interesting and important scientific question. However, as far as the authors know,
there are few studies focusing on the influence of seasonal variations in stratification on internal
tidal energy flux in the SCS, probably due to limited observations in the deep SCS. To explore
the potential influence of seasonal stratification on ITs' energy flux in the SCS, we investigate
the seasonal variations of abyssal stratification (averaged below 2000 m) and tidal energy flux
at the Luzon Strait, which is the main generation site of ITs in the SCS. As shown in **Fig. R11**,
the abyssal stratification and ITs' energy flux exhibit a similar seasonal variation, strengthening
in summer and autumn while weakening in winter and spring. The maximum values for both
stratification and energy flux appear in summer, showing a 15% and 23% increase, respectively,

compared to the minimum values appearing in winter. Notably, the strengthening of ITs during
summer and autumn is temporally consistent with the peak season of typhoons in the SCS
(Wang et al. 2007), heightening the suppressing effect of ITs on SCS TCs.

It should be noted that here we only explore the seasonal variations of the abyssal stratification
and ITs' energy flux at the Luzon Strait. Actually, the impact of monsoon flow on abyssal
stratification and consequently internal tidal energy flux remains inconclusive. Moreover, the
seasonal variations of the upper ocean stratification can influence the seasonal cycle of ITs'
energy flux by modulating the propagation and vertical modes of ITs. Therefore, although not
the specific topic of this study, the relationship among the monsoon flow, seasonal variations
of ocean circulation and stratification, and hence ITs' energy flux as well as their suppressing
effect on TCs, is a complicated question, and needs to be investigated in future studies.

Following your suggestion, we have added a discussion on the potential influences of monsoon
flow in the revised manuscript (Lines 236–240 in the clean version).

**Fig. R11. Stratification and internal tidal energy flux at the Luzon Strait. a** Topography in the SCS.

The red box represents the chosen region for calculation. **b** Seasonal variation of stratification averaged
below 2000 m, using temperature/salinity dataset from the World Ocean Atlas 2018 (WOA18)
dataset. **c** Seasonal variation of area-integrated internal tidal energy flux at the Luzon Strait. The
calculation is based on data from the MITgcm LLC4320 simulation (with a horizontal resolution of
$1/48^\circ$ with 90 vertical layers over the globe, for 14 months from Sep. 2011 to Nov. 2012) by the ECCO
team (Data source: ECCO Data Portal (<https://data.nas.nasa.gov/ecco/data.php>)).

2. Does the TC-IT interaction depend on the TC track? The IT effects intensify the oceanic
current evidently; what about the contribution of variation in mixed layer depth and barrier
layer thickness?

Generally, internal tides (ITs) in the SCS are not spatially homogeneous, manifesting stronger
energy in the north SCS and weaker energy in the south SCS (Zhao 2014; Xu et al. 2016).
Stronger ITs in the north SCS potentially enable more energy supply to enhance the TC-
induced cooling than the south SCS. Therefore, the TC-IT interaction may depend on the
spatial distribution of the TC track, that is, TCs passing over regions with strong or weak
background ITs.

To examine the dependence of TC-IT interaction on TC tracks passing over regions with strong
or weak ITs, we compare ITs and TC-induced SST cooling effect between the north and south
SCS. The strength of ITs between the north and south SCS is compared using two moorings
(**Fig. R12a**). Results show that ITs' energy in the north SCS is about six times larger than that
in the south SCS (**Fig. R12b**), confirming results from satellite observations (Zhao 2014) and
numerical simulations (Xu et al. 2016; Zhao et al. 2019). The difference in average SST cooling
induced by intense TCs between the north and south SCS is up to 0.5°C (-3.4°C in the north
SCS vs. -2.9°C in the south SCS) (**Fig. R12c**), suggesting that stronger background ITs in the
north SCS result in larger SST cooling. Correspondingly, the amplitude of negative TC
intensification rate in the north SCS ($-13.2\text{ m s}^{-1}\text{ day}^{-1}$) is 47% larger than that in the south
SCS ($-9.0\text{ m s}^{-1}\text{ day}^{-1}$) for intense TCs.

Therefore, TC-IT interaction indeed varies with the locations of TC track because the strength
of ITs is spatially inhomogeneous in the SCS. TC-IT interaction is more effective when TC
passes over the north SCS where ITs are stronger than the south SCS. The related discussion
has been added in the revised manuscript (Lines 228–236 in the clean version).

**Fig. R12. Comparison of IT effect in the north and south SCS.** **a** Locations of two moorings in the
 north and south SCS. Mooring NSCS (red star; 21.1°N, 117.9°E) is located in the north SCS (north of
 17°N). Mooring SSCS (blue star; 13.0°N, 112.9°E) is located in the south SCS (south of 17°N).
 Background color represents the topography of the SCS. The 1000 m isobath is shown as black dashed
 contours. The boundary (17°N) of south and north SCS is indicated by the magenta dashed line. **b**
 Average power spectrum of the horizontal currents of internal waves in the north (red) and south (blue)
 SCS. The unit of x-axis is cpd (cycles per day). The frequencies of near-inertial waves (f), background
 diurnal (D1), and semidiurnal (D2) ITs are indicated by vertical dashed lines. The power spectrum is
 derived from synchronous ocean current observations at the two moorings from July 2016 to July 2017.
 **c** TC-induced SST cooling and TC intensification rate for intense TCs in the north and south SCS,
 derived from the composite results of satellite observations.

As the reviewer pointed out, variations of mixed layer depth and barrier layer thickness can
 also modulate the TC-induced cooling effect. Shallow (deep) mixed layer usually corresponds
 to a strong (weak) thermal stratification in the upper ocean. When passing over oceans with a
 shallower mixed layer and stronger upper ocean thermal stratification, TCs entrain subsurface
 cold water more easily and thus usually induce a stronger SST cooling, than regions with a
 deeper mixed layer (Wang et al. 2016). The SCS subsurface thermal stratification is relatively
 strong among global oceans (Mei et al. 2015; Sun et al. 2017), but is weaker than that in the

Eastern Pacific (**Figs 3c** and **3d**; Supplementary **Fig. S4**). Despite the weaker stratification, the
SCS still experiences 50–100% stronger cooling than the Eastern Pacific for TCs with equal
attributes (**Fig. 2e**; Supplementary **Fig. S5**). This indicates that the mixed layer is not the cause
for the strongest cooling in the SCS.

Due to enhanced saline stratification, barrier layer can suppress the TC-induced vertical mixing
and thus reduce the magnitude of SST cooling during TC passage, which usually favors the
intensification of TCs (Balaguru et al. 2012). The barrier layer in the SCS has substantial
seasonal variations; it is thicker and occurs more frequently in summer and early autumn, when
SCS TCs are most active (Zeng & Wang 2017). Therefore, the presence of thick and frequent
barrier layer in TC season, cannot be the reason for the strongest SST cooling in the SCS.

3. What about the islands' effects on the weak TC intensification over the SCS? The islands
may increase the surface roughness and decrease the TC intensification.

To avoid the island's effects, originally we have already excluded TC track points located
within 100 km of the land or landfall within 24 hours before composite analysis. Therefore, the
lowest TC intensification rate observed in the SCS is not subject to the islands' effects from
increased surface roughness.

In addition to the possible effect of increased surface roughness, another potential effect related
to the high mountains in Philippine islands is also ruled out. Specifically, passing over the
Philippine mountains might compromise TC structure and indirectly influence the subsequent
TC intensification after passing the mountain (Brand & Blelloch 1974; Chou et al. 2011). To
examine the potential influence of mountain passing, we separate SCS TC into two groups
(with and without mountain passing) and compare the TC intensification rates (**Figs R13a** and
**R13b**). The difference in intensification rates between the two groups is small, and even the
intensification rate of TCs without mountain passing is lower than that of TCs with mountain
passing for category 2–4 TCs (**Fig. R13c**). Furthermore, the intensification rates consistently
remain lower than the global average regardless of whether TCs pass over the Philippine
mountains or not. Therefore, the passage over the Philippine mountains may not be the cause
for the weak intensification of SCS TCs. We describe this at Lines 116–118 in the manuscript
(clean version).

Considering the percentage of intense TCs might also be related to the small basin size of the
 SCS, we perform a ‘fixed-window’ analysis, which compares the percentages calculated when
 TCs in all oceans travel as long a distance as those in the SCS. Results show when using the
 ‘fixed-window’ analysis, which excludes the impact of basin size, the SCS percentage of
 intense TCs is still the lowest (**Fig. R14**). We describe the effect of islands, passage over the
 Philippine mountain, and basin size at Lines 105–120 in the manuscript (clean version).

 **Fig. R13 (Supplementary Fig. S2). Tracks and intensification rates (IRs) for TCs that enter the**
 **SCS passing over the Philippine mountains (POPM) or not (NonPOPM). a** Tracks of POPM subset
 (217 TCs). **b** Tracks of NonPOPM subset (184 TCs). **c** Average IRs versus TC category for all SCS
 TCs (green), POPM subset (red) and NonPOPM (blue) subset, as well as the average IR in other five
 TC-rich oceans (black). To increase sampling number, all TC track points not within 100 km of land
 are used here to calculate IRs in (c). The error bars in (c) indicate the 90% confidence interval.

**Fig. R14 (Supplementary Fig. S1). Percentage of intense TCs from the ‘fixed-window’ analysis.**

The analysis compares the percentage of intense TCs, calculated for TCs in all oceans travelling the
same distance as those in the SCS. **a** Result from the TC-case based method. **b** Result from the TC
track-point based. To avoid interference, TCs generated in the western North Pacific and translated into
the SCS were first pre-excluded, i.e., only TCs locally generated in the SCS are considered for the SCS
analysis. For TCs of the other five TC-active ocean basins, we only account for the segment from
genesis to a traveling distance from 700 to 1518 km for each TC. Thus, the relatively small basin size
cannot explain the lowest percentage of intense TCs in the SCS.

4. Have you checked more cases to ascertain the differences in TCs between SCS and North
Atlantic? Why not compare the TCs in the SCS and northern IO since they have more similar
monsoon climate backgrounds?

Thanks for the constructive comments. Following your suggestion, we search in-situ
observations from global mooring/buoy databases and conducted comparisons based on more
TC cases; the results based on these cases are generally consistent with those shown in the
original manuscript. The mooring/buoy observations for comparison should have upper ocean

current observations and should be subject to similar TC characteristics. Based on the South
China Sea Mooring Array (SCSMA) constructed by the Ocean University of China since 2009,
we obtain hundreds of cases where moorings are located within 200 km from the TC center in
the SCS (**Figs R15a** and **R15c**). In other TC-active ocean basins, available mooring or buoy
observations with subsurface current information during TC passages are rare. We obtain three
buoys (42002, 42038, and 42041) with upper ocean current observations in the North Atlantic
(NA) from the National Data Buoy Center. The corresponding NA TC tracks and buoys,
located within 200 km from the TC center, are shown in **Figs R15b** and **R15d**. In total, 14
cases in which buoys record the upper ocean current responses to NA TCs are identified.

Based on the TC-mooring/buoy cases shown in **Fig. R15**, more than 10 pairs of cases are
identified (**Table R2**) and four pairs are shown here (**Figs R16–R19**). Consistent with results
in **Fig. 4**, the upper ocean responses to SCS TCs (**Figs R16–R19**) show more complex and
stronger currents and shears than responses to NA TCs. Power spectra of upper ocean currents
during SCS TCs show energetic near-inertial waves (NIWs) generated by TCs and also
background ITs ($D1$, $D2$). In addition, energetic spectral peaks also appear at frequencies of
secondary waves ($fD1$, $fD2$, $D2f$), indicating the nonlinear coupling between TC-generated
NIWs and background ITs. However, the power spectra of upper ocean currents during NA
TCs only show energetic peaks of TC-generated NIWs. On average, the integrated internal
wave energy during SCS TCs is more than five times larger than that during NA TCs, which
can effectively enhance the upper ocean mixing and enlarge SST cooling during TCs.

Overall, based on observations from these new cases, the interactions between background ITs
and TC-generated NIWs amplify turbulent entrainment and thus enlarge the SST cooling.

[REDACTED]

**Fig. R15.** [REDACTED]

**Table R2.** TC information in the comparable TC cases between the South China Sea (SCS) and the

North Atlantic (NA).

Ocean basin	TC name	Wind speed (m s ⁻¹)	Translation speed (m s ⁻¹)	Distance (km)
SCS	Kompasu (2021)	28.3	6.8	222
NA	Dolly (2008)	30.9	4.9	231
SCS	Haitang (2017)	20.6	7.6	-145
NA	Ivan (2004)	18.0	7.7	-144
SCS	Atsani (2020)	23.1	4.4	-65
NA	Matthew (2004)	20.6	4.1	-88

SCS	Hato (2017)	25.7	6.6	127
NA	Bill (2003)	23.1	6.0	102
SCS	Cimaron (2013)	20.6	8.7	142
NA	Erika (2003)	23.1	9.8	156
SCS	Thirteen (2018)	18.0	6.0	-144
NA	Ivan (2004)	18.0	7.7	-144
SCS	Danas (2019)	18.0	7.8	-144
NA	Ivan (2004)	18.0	7.7	-144
SCS	Barijat (2018)	20.6	4.4	-90
NA	Matthew (2004)	20.6	4.1	-88
SCS	Choi-wan (2021)	18.0	4.8	-88
NA	Matthew (2004)	20.6	4.1	-88
SCS	Barijat (2018)	20.6	4.4	95
NA	Bill (2003)	23.1	6.0	102
SCS	Mekkhala (2020)	23.1	5.9	102
NA	Bill (2003)	23.1	6.0	102

* 'Wind speed' and 'translation speed' mean the maximum wind speed and translation speed of the TC
when TC passes the mooring/buoy. 'Distance' represents the distance between the TC center and
mooring/buoy, and positive (negative) sign means the mooring/buoy is located to the right (left) of the
TC track.

Fig. R16. Comparisons of upper ocean response to SCS TC (with ITs) and NA TC (without ITs). **a** TC track and mooring position (red star) in the SCS. **b** TC track and buoy position (blue star) in the NA. The time when TC passed mooring (buoy) is written in the bottom left corner of **(a)** and **(b)**. **c** Mooring-observed ocean horizontal currents during SCS TC passage. **d** Buoy-observed ocean horizontal currents during NA TC passage. The currents in **(c)** and **(d)** are 48 h high-pass filtered with a fourth-order Butterworth filter. **e** Power spectrum density (PSD) of horizontal currents along fixed depths during and after the passage of SCS TC (red) and NA TC (blue). The PSD is averaged over the upper 150 m. **f** Power spectrum density (PSD) of horizontal currents along isopycnals during and after the passage of SCS TC (red) and NA TC (blue). The PSD in the SCS is averaged over isopycnals marked by dark lines in **(c)**. Due to the lack of temperature observations, the PSD in the NA is still averaged over the upper 150 m.

**Fig. R17.** Same as **Fig. R16**, but for a different pair of TC cases.

**Fig. R18.** Same as **Fig. R16**, but for a different pair of TC cases.

**Fig. R19.** Same as **Fig. R16**, but for a different pair of TC cases.

We agree with you that it's a good point to compare TCs in the SCS and northern Indian Ocean (NIO) since they have more similar monsoon climate backgrounds. In the original manuscript, due to the relatively low frequency of TCs in the NIO, we compare TCs between the SCS and the Indian Ocean (NIO plus South Indian Ocean (SIO)) rather than the NIO alone. Following your suggestion, here we further make a comparison of TCs between the SCS and the NIO or SIO (**Fig. R20**). Consistent with results in **Fig. 2**, although having similar monsoon climate backgrounds, in the SCS the SST cooling is stronger and TC intensification rate is lower than that in the NIO. The results associated with the comparison between the SCS and SIO are similar. Therefore, these results further demonstrate that the SCS indeed has the strongest TC-induced SST cooling and the lowest TC intensification rate among all global TC-active basins.

**Fig. R20. Comparisons of TC-induced SST cooling between the SCS, the north Indian Ocean**
 **(NIO), and the south Indian Ocean (SIO). a** TC intensification rate versus TC category. **b** TC-induced
 SST cooling versus TC category.

5. How to understand the stronger oceanic current but weaker TC-related winds? Because the
 entrainment cooling is sensitive to the surface winds.

Thanks for the thoughtful comment. As you pointed out, mooring observations show stronger
 ocean currents during TC passages in the SCS, although TCs in the SCS are relatively weaker.
 The stronger oceanic current in the SCS is a result of the superposition and interaction between
 internal tides-related currents and TC-induced near-inertial currents (Guan et al. 2014; Liu et
 al. 2018). Therefore, although TC-related winds and corresponding wind-driven currents are
 relatively weak, the total ocean currents in the SCS are stronger owing to the presence of strong
 background internal tides.

The strength of turbulent mixing and entrainment is determined by the shear instability of
 horizontal ocean currents. The interactions between background internal tides and TC-
 generated oceanic near-inertial waves can effectively amplify the shear instability and thus the
 turbulent entrainment in the upper ocean (Guan et al. 2014; Liu et al. 2018) and consequently
 heighten the SST cooling (**Fig. 5**). Therefore, the TC-related entrainment cooling in the SCS,
 which hosts powerful internal tides, is larger than that in other TC-active oceans.

**References:**

Balaguru, K. et al. Ocean barrier layers' effect on tropical cyclone intensification. *Proc. Natl.*
*Acad. Sci. U.S.A.* **109** (36), 14343-14347 (2012).

Brand, S. & Blelloch, J. W. Changes in the characteristics of typhoons crossing the island of
Taiwan. *Mon. Wea. Rev.* **102**(10), 708-713 (1974).

Chou, K. H., Wu, C. C. & Wang, Y. Eyewall Evolution of Typhoons Crossing the Philippines
and Taiwan: An Observational Study. *Terrestrial, Atmospheric & Oceanic Sciences* **22**(6)
(2011).

Guan, S., Zhao, W., Huthnance, J., Tian J. & Wang, J. Observed upper ocean response to
typhoon Megi (2010) in the Northern South China Sea. *J. Geophys. Res.* **119**, 3134-3157 (2014).

Liu, J., He, Y., Li, J., Cai, S., Wang, D. & Huang, Y. Cases study of nonlinear interaction
between near-inertial waves induced by typhoon and diurnal tides near the Xisha Islands. *J.*
*Geophys. Res.* **123**, 2768-2784 (2018).

Mei, W., Lien, C. C., Lin, I. I. & Xie, S. P. Tropical cyclone-induced ocean response: A
comparative study of the South China Sea and tropical northwest Pacific. *J. Clim.* **28**(15), 5952-
5968 (2015).

Müller, M. On the space-and time-dependence of barotropic-to-baroclinic tidal energy
conversion. *Ocean Modelling* **72**, 242-252 (2013).

Sun, J., Oey, L. Y., Xu, F. & Lin, Y. C. Sea level rise, surface warming, and the weakened
buffering ability of South China Sea to strong typhoons in recent decades. *Sci. Rep.* **7**(1) (2017).

Wang, B., Huang, F., Wu, Z., Yang, J., Fu, X. & Kikuchi, K. Multi-scale climate variability of
the South China Sea monsoon: A review. *Dynamics of Atmospheres and Oceans* **47**(1-3), 15-
37 (2009).

Wang G., Su J., Ding Y. & Chen D. Tropical cyclones genesis over the South China Sea.
*Journal of Marine Systems* **68**(3-4), 318-326 (2007).

Wang, G., Wu, L., Johnson, N. C. & Ling, Z. Observed three-dimensional structure of ocean
cooling induced by Pacific tropical cyclones. *Geophys. Res. Lett.* **43**(14), 7632-7638 (2016).

Xu, Z., Liu, K., Yin, B., Zhao, Z., Wang, Y. & Li, Q. Long-range propagation and associated
variability of internal tides in the South China Sea. *J. Geophys. Res.* **121**(11), 8268-8286 (2016).

Zeng, L. & Wang, D. Seasonal variations in the barrier layer in the South China Sea:
characteristics, mechanisms and impact of warming. *Clim. Dyn.* **48**, 1911–1930 (2017).

Zhao, Z. Internal tide radiation from the Luzon Strait. *J. Geophys. Res.* **119**, 5434-5448 (2014).

Zhao, Z., Wang, J., Menemenlis, D., Fu, L. L., Chen, S. & Qiu, B. Decomposition of the
multimodal multidirectional M2 internal tide field. *Journal of Atmospheric and Oceanic*
*Technology* **36**(6), 1157-1173 (2019).

Zu, T. et al. Interannual variation of the South China Sea circulation during winter: intensified
in the southern basin. *Clim. Dyn.* **52**, 1917-1933 (2019).

REVIEWERS' COMMENTS

Reviewer #3 (Remarks to the Author):

I appreciate the authors' careful revision according to the reviewers' comments. I think this version can be accepted to publish.

**Authors' response to the second-round reviewers' comments**

Ref. Number: NCOMMS-23-37157A

Title: Ocean Internal Tides Suppress Tropical Cyclones in the South China Sea

**REVIEWERS' COMMENTS**

Reviewer #3 (Remarks to the Author):

I appreciate the authors' careful revision according to the reviewers' comments. I think this
version can be accepted to publish.

We thank the reviewer for his/her constructive suggestions and comments for significantly
improving the manuscript through the review process.
